# FedCDWA: Decoupled Federated Prototype Distillation with Hierarchical Wasserstein Aggregation

**Zhenshen Liu** [1]   **Kai Fan** [1]   **Wenjie Li** [2]   **Kuan Zhang** [3]   **Hui Li** [1]   **Yintang Yang** [4]

## Abstract

Federated learning enables decentralized clients to collaboratively train models without sharing local data. However, heterogeneous client distributions often induce client drift and hinder convergence. This paper proposes FedCDWA, a decoupled hierarchical federated prototype distillation framework. FedCDWA decouples client-side personalized distillation from server-side mutual distillation to mitigate distillation-induced optimization conflicts. It further adopts Hierarchical Wasserstein Aggregation to aggregate prototypes without restrictive parametric assumptions while preserving intra-class structure and inter-class geometry. To achieve finer-grained feature alignment, Prototype–Variance Dual Alignment matches feature means and variances in the feature space. We prove convergence guarantees for FedCDWA. Experiments on three datasets demonstrate that FedCDWA consistently improves both global and personalized accuracy across heterogeneity levels, with smaller performance degradation under more severe heterogeneity.

## 1. Introduction

Federated learning (FL) (McMahan et al., 2017) has emerged as a highly promising distributed learning paradigm that enables collaborative model training across decentralized clients while preserving data privacy. This paradigm has shown great potential for application in sensitive domains such as finance and healthcare (Ma et al., 2021; Bonawitz et al., 2019). However, in practical deployments, client data distributions are typically non-independent and identically distributed (Non-IID) (Zhao et al., 2018; Li et al., 2021c; Zhang et al., 2023), where both label and feature distributions can vary significantly across clients. Such statistical heterogeneity gives rise to severe client drift (Karimireddy et al., 2020; Jiang et al., 2024; Reddi et al., 2021), causing local updates to deviate markedly from the global objective, which slows convergence and degrades the generalization ability of the global model.

To mitigate data heterogeneity, existing work has evolved primarily in three directions. Optimization-centric strategies, typified by FedProx (Li et al., 2020) and SCAFFOLD (Karimireddy et al., 2020), attempt to correct local gradients to better align them with the global objective via proximal terms or control variates. Alternatively, personalization frameworks such as FedPer (Arivazhagan et al., 2019) decouple global representations from local classifiers to accommodate client-specific distribution shifts. More recently, prototype- and distillation-based approaches, notably FedProto (Tan et al., 2022a), FedHKD (Chen et al., 2023), and FedGMKD (Zhang et al., 2024), have garnered significant attention by leveraging class-level feature sharing to enhance knowledge transfer.

Despite these advances, federated prototype distillation approaches still face three significant challenges under non-IID settings, as illustrated in Fig. 1. First, coupled bidirectional distillation may cause gradient conflicts (Lee et al., 2022). Clients distill from server-issued soft targets, while the server refines the global model using distillation objectives constructed from client-returned knowledge within the same communication round. This coupling forms a feedback loop between teacher targets and model updates, making the targets non-stationary and prone to gradient conflicts. Second, server-side prototype fusion often relies on simple aggregation (Tan et al., 2022a; Chen et al., 2023). Under highly skewed client distributions, global prototypes can be biased toward dominant clients, thereby distorting the geometry of the global feature space. Third, output-space point-wise alignment reduces each class to a single centroid (Tan et al., 2022a; Chen et al., 2023; Zhang et al.,

---

[1] State Key Laboratory of Integrated Service Networks, School of Cyber Engineering, Xidian University, Xi'an, Shaanxi 710071, China  [2] School of Computer and Cyber Security, Hebei Normal University, Shijiazhuang 050024, China  [3] Department of Electrical and Computer Engineering, University of Nebraska–Lincoln, Lincoln, NE 68588, USA  [4] Key Laboratory of the Ministry of Education for Wide Band Gap Semiconductor Materials and Devices, Xidian University, Xi'an, Shaanxi 710071, China . Correspondence to: Kai Fan <kfan@mail.xidian.edu.cn>.

*Proceedings of the 43rd International Conference on Machine Learning*, Seoul, South Korea. PMLR 306, 2026. Copyright 2026 by the author(s).

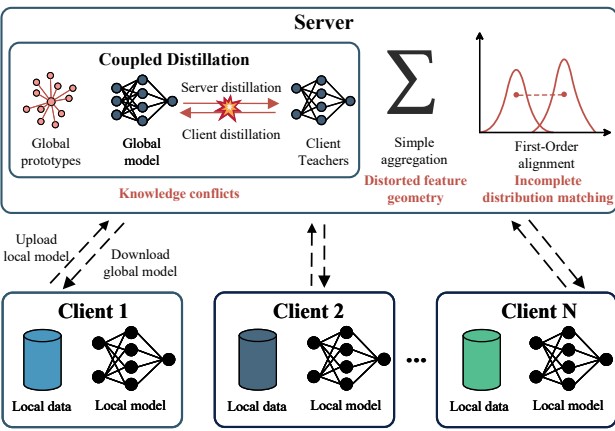

*Figure 1.* Challenges in federated prototype distillation. Coupled distillation causes knowledge (gradient) conflicts. Simple aggregation distorts geometry. First-order alignment yields mismatch, hindering knowledge transfer. FedCDWA addresses these via DDS, HWA, and PVDA for robust generalization.

2024). It overlooks cross-client differences in intra-class dispersion and uncertainty. Consequently, a "same center, different shape" mismatch may persist, weakening robustness and generalization.

To address these challenges, we propose FedCDWA, a decoupled hierarchical federated prototype distillation framework. First, we propose a Decoupled Distillation Strategy (DDS) to resolve optimization conflicts induced by coupled bidirectional distillation. On the client side, DDS uses the local classifier to infer personalized soft predictions from global prototypes for adaptive knowledge absorption. The server-provided teacher targets are treated as fixed and are therefore unaffected by client-side personalization. On the server side, mutual distillation is employed to further refine the global model. Second, to overcome the limitations of parametric aggregation, we propose Hierarchical Wasserstein Aggregation (HWA). Guided by optimal transport theory, HWA aggregates prototypes both within and across classes. It preserves the underlying geometric structure of the feature space without relying on restrictive distributional assumptions. Third, we design a Prototype–Variance Dual Alignment (PVDA) mechanism to address incomplete distribution alignment. This mechanism jointly matches the first- and second-order statistics of features. By integrating DDS, HWA, and PVDA, FedCDWA achieves stable knowledge transfer, preserves geometric structure, and enables fine-grained distribution matching. This mitigates client drift and enhances generalization under heterogeneous data distributions.

The main contributions of this paper are summarized as follows:

- We propose FedCDWA, a novel framework that effec-

tively resolves gradient conflicts via a Decoupled Distillation Strategy (DDS). By separating personalized inference from mutual distillation, FedCDWA achieves stable and efficient knowledge transfer under Non-IID settings.

- We design the Hierarchical Wasserstein Aggregation (HWA) and Prototype–Variance Dual Alignment (PVDA) mechanism. HWA leverages optimal transport theory to preserve the geometric integrity of prototypes without relying on parametric assumptions; meanwhile, the PVDA mechanism jointly calibrates first- and second-order statistics to achieve superior feature alignment.

- A rigorous convergence analysis of FedCDWA is provided, quantifying the trade-off between local personalization and global consistency.

- Extensive experiments on three datasets show that FedCDWA surpasses state-of-the-art (SOTA) methods in both global and local accuracy, while exhibiting strong robustness across varying degrees of data heterogeneity.

## 2. Related Work

### 2.1. Federated Learning under Data Heterogeneity

Data heterogeneity across clients poses a fundamental challenge in FL. It induces client drift and degrades convergence. FedAvg (McMahan et al., 2017) performs simple parameter averaging. It works well under IID data but suffers under Non-IID settings. To mitigate client drift caused by heterogeneous local objectives, FedProx (Li et al., 2020) adds a proximal regularizer to constrain local updates. SCAF-FOLD (Karimireddy et al., 2020) uses control variates to correct client update directions. FedNova (Wang et al., 2020) eliminates weight bias through gradient normalization. FedDyn (Acar et al., 2021) refines aggregation with a dynamic regularizer. Alternatively, prompt-based methods address heterogeneity via decoupled client distillation. PEPSY (Nguyen et al., 2025) reconfigures client embeddings for missing-data heterogeneity. FED-PRIME (Phung et al., 2025) decouples multimodal prompt optimization from server aggregation. Despite progress at the parameter level, these methods focus primarily on model parameter alignment and overlook the feature-level distributional differences across clients. This limits their effectiveness under severely skewed data distributions where clients exhibit significant differences in feature distributions.

### 2.2. Personalized Federated Learning

To address the limitations of global model approaches, personalized FL adapts to heterogeneous data by learning

customized models for individual clients. Architecture-decoupling methods split models into shared and private components. FedPer (Arivazhagan et al., 2019) aggregates only the feature extractor while keeping classifiers local. Ditto (Li et al., 2021b) maintains both a global and a local model for each client and uses a regularization term to balance global consistency and local adaptability. Cluster-based approaches group similar clients. IFCA (Ghosh et al., 2020) and FeSEM (Xie et al., 2023) cluster clients by loss or gradient similarity. Meta-learning methods like Per-FedAvg (Fallah et al., 2020) learn initialization parameters for rapid local adaptation. However, these methods rely on architectural decomposition or parameter-space fine-tuning. They make limited use of prototype-level knowledge sharing across clients. This restricts the efficiency of knowledge transfer.

### 2.3. Prototype and Knowledge Distillation in FL

To improve knowledge transfer efficiency, prototype learning and knowledge distillation enable efficient knowledge sharing under communication constraints by transmitting compact representations instead of full model parameters. FedProto (Tan et al., 2022a) achieves class-level knowledge sharing via prototype aggregation. FedPCL (Tan et al., 2022b) introduces prototype-wise contrastive learning with pre-trained extractors. FedGMKD (Zhang et al., 2024) aggregates class prototypes using Gaussian mixture models with discrepancy-aware weighting to capture distributional characteristics. However, these methods rely on restrictive parametric assumptions or reduce classes to single centroids, struggling to capture complex feature manifolds.

Distillation-based methods transfer knowledge through soft targets. FedMD (Li & Wang, 2019) enables knowledge transfer via consensus distillation on a public dataset. Fed-Gen (Zhu et al., 2021) uses a generator to synthesize data for knowledge distillation without requiring a public dataset. FedKD (Wu et al., 2022) transfers knowledge from the global model to local models. FedHKD (Chen et al., 2023) proposes hyper-knowledge distillation for improved local performance. However, these approaches face key challenges: coupled distillation induces gradient conflicts (Lee et al., 2022), and overlooks intra-class distributional differences across clients (Wang et al., 2024).

## 3. Methodology

### 3.1. Problem Formulation

In FL, the system includes $n$ clients, where client $i$ holds a local private dataset $\mathcal{D}_i = \{(\mathbf{x}_j^i, y_j^i)\}_{j=1}^{|\mathcal{D}_i|}$. In practice, client data are typically heterogeneous, with label and feature distributions differing significantly across clients. The goal

is to minimize the global loss:

$$\min_{\theta} \mathcal{L}_{\text{glob}}(\theta) = \sum_{i=1}^{n} \rho_i \mathcal{L}_i(\theta, \mathcal{D}_i), \quad (1)$$

where $\mathcal{L}_i(\theta, \mathcal{D}_i)$ denotes the local loss of client $i$, $\rho_i = \frac{|\mathcal{D}_i|}{N_{\text{total}}}$ is the aggregation weight of client $i$, with $N_{\text{total}} = \sum_{j=1}^{n} |\mathcal{D}_j|$ denoting the total number of samples across all clients, such that $\sum_{i=1}^{n} \rho_i = 1$. The local loss is defined as:

$$\mathcal{L}_i(\theta, \mathcal{D}_i) = \frac{1}{|\mathcal{D}_i|} \sum_{(\mathbf{x}, y) \in \mathcal{D}_i} \ell(\theta; \mathbf{x}, y), \quad (2)$$

where $\ell(\theta; \mathbf{x}, y)$ denotes the per-sample loss on $(\mathbf{x}, y)$ and $\theta$ denotes the model parameters. As a classical baseline, FedAvg (McMahan et al., 2017) updates the global model by a weighted average of client parameters:

$$\theta_g^t = \sum_{i=1}^{n} \rho_i \theta_i^{t-1}, \quad (3)$$

where $\theta_g^t$ denotes the global model parameters at round $t$, and $\theta_i^{t-1}$ denotes the local model parameters of client $i$ at round $t-1$. In non-IID settings, directly aggregating parameters trained on heterogeneous data often leads to client drift, thereby degrading the performance of the global model.

To address this challenge, FedHKD (Chen et al., 2023) improves personalized models through hyper-knowledge distillation. Building upon this, FedGMKD (Zhang et al., 2024) further enhances global models' performance by introducing Cluster Knowledge Fusion and Discrepancy-Aware Aggregation based on Gaussian Mixture Models. However, these approaches still face limitations in handling gradient conflicts, preserving geometric structure, and capturing intra-class variance. Inspired by these insights, we propose FedCDWA, which integrates a DDS, HWA, and PVDA to achieve robust knowledge transfer and fine-grained distribution matching. Detailed workflow is provided in Appendix A.

### 3.2. Decoupled Distillation Strategy

Existing federated distillation methods employ coupled bidirectional distillation, where clients and server act as both teachers and students. Under non-IID data, this coupling induces gradient conflicts (Lee et al., 2022) and ignores client-specific distribution shifts (Chen et al., 2023). To address this, we propose DDS. DDS separates knowledge transfer into two independent stages: client-side personalized prototype distillation and server-side mutual distillation. This decoupling eliminates gradient conflicts while enabling adaptive knowledge absorption.

### 3.2.1. CLIENT-SIDE PERSONALIZED PROTOTYPE DISTILLATION

In federated image classification, the model parameters $\theta_i$ of each client $i$ are decomposed into a feature extractor $f_i(\cdot; \theta_i^{fe})$ and a classifier $h_i(\cdot; \theta_i^{cls})$. For each mini-batch $(\mathbf{X}, \mathbf{y}) \sim \mathcal{D}_i$, the model performs inference via the feature extractor and the classifier:

$$
\begin{aligned}
\mathbf{F}_i &= f_i(\mathbf{X}; \theta_i^{\text{fe}}), \\
\mathbf{Z}_i &= h_i(\mathbf{F}_i; \theta_i^{\text{cls}}), \\
\mathbf{P}_i &= \text{softmax}(\mathbf{Z}_i/T_c),
\end{aligned} \quad (4)
$$

where $\mathbf{F}_i$ is the feature matrix, $\mathbf{Z}_i$ denotes the logits, $\mathbf{P}_i$ denotes the soft predictions, and $T_c$ is the temperature parameter.

We define $\mathcal{C}_i$ as the set of classes available at client $i$, and $\mathcal{D}_i^c$ as the sample set of class $c$ at client $i$. For each class $c \in \mathcal{C}_i$, client $i$ computes the local soft prediction $\boldsymbol{p}_i^c$, defined as:

$$
\boldsymbol{p}_i^c = \frac{1}{|\mathcal{D}_i^c|} \sum_{(\mathbf{x},y)\in\mathcal{D}_i^c} \text{softmax}(h_i(f_i(\mathbf{x}; \theta_i^{fe}); \theta_i^{cls})/T_c),
$$
(5)

**Personalized Prototype Inference.** At round $t$ ($t \geq 2$), client $i$ receives from the server the global prototypes $\{\boldsymbol{\mu}_g^c\}$ and the global soft predictions $\{\boldsymbol{p}_g^c\}$. Rather than passively mimicking the global predictions, the client uses its local classifier $h_i$ to infer personalized soft predictions from the global prototypes:

$$
\mathbf{Z}_{\text{proto}} = h_i(\{\boldsymbol{\mu}_g^c\}; \theta_i^{cls}), \quad \mathbf{Q}_{\text{proto}} = \text{softmax}(\mathbf{Z}_{\text{proto}}/T_c).
$$
(6)

where $\mathbf{Z}_{\text{proto}}$ is the logits matrix derived from global prototypes, $\mathbf{Q}_{\text{proto}}$ is the corresponding soft-prediction matrix, and $\mathbf{Q}_{\text{proto}}[c,:]$ represents client $i$'s personalized prediction for the global prototype of class $c$. Since $h_i$ has already adapted to the local distribution $\mathcal{D}_i$, this mechanism enables different clients to extract differentiated, locally adapted knowledge from the same global prototypes, thereby facilitating personalized knowledge absorption.

**Decoupled Distillation Objective.** The client's training objective is defined as:

$$
\mathcal{L}_c(\theta_i) = \begin{cases} \frac{1}{|\mathcal{D}_i|} \sum_{(\mathbf{x},y)\in\mathcal{D}_i} \ell(h_i(f_i(\mathbf{x}; \theta_i^{fe}); \theta_i^{cls}), y), & t = 1 \\ \frac{1}{|\mathcal{D}_i|} \sum_{(\mathbf{x},y)\in\mathcal{D}_i} \ell(h_i(f_i(\mathbf{x}; \theta_i^{fe}); \theta_i^{cls}), y) \\ \quad + \frac{\xi}{|\mathcal{C}_g|} \sum_{c\in\mathcal{C}_g} \text{KL}\left(\mathbf{Q}_{\text{proto}}[c,:] \parallel \boldsymbol{p}_g^c\right), & t \geq 2 \end{cases}
$$
(7)

where $\ell(\cdot, \cdot)$ is the cross-entropy (CE) loss and $\xi$ is the weight of the distillation loss. In the first round ($t = 1$), the server has not yet aggregated any client knowledge, so $\{\boldsymbol{\mu}_g^c, \boldsymbol{\sigma}_g^c, \boldsymbol{p}_g^c\} = \emptyset$, and the client is trained using only

the CE loss. Starting from the second round ($t \geq 2$), the client uses the full loss function, including CE and one-way distillation losses, quantified by the Kullback-Leibler (KL) divergence.

Notably, in the distillation term, the global prediction $\boldsymbol{p}_g^c$ is treated as a fixed target. Therefore, the distillation loss updates only the local parameters $\theta_i$, and its gradients do not flow to the server-side knowledge, effectively avoiding gradient conflicts and ensuring that clients can stably absorb global knowledge while maintaining personalization.

### 3.2.2. SERVER-SIDE BIDIRECTIONAL MUTUAL DISTILLATION

After local training is completed, the server aggregates the client parameters into $\theta_g^t$ and aggregates the local prototypes into $\boldsymbol{\mu}_g^c$ (see Section 3.3). Subsequently, the server refines the global model using the aggregated knowledge.

Using the updated global model $\theta_g^t$, the server performs inference on the global prototypes:

$$
\boldsymbol{z}_g^c = h_g(\boldsymbol{\mu}_g^c; \theta_g^{cls,t}), \quad \boldsymbol{p}_g^c = \text{softmax}(\boldsymbol{z}_g^c/T_s), \quad (8)
$$

Here, $\boldsymbol{z}_g^c$ denotes the logits of the global model on the prototype of class $c$, and $\boldsymbol{p}_g^c$ denotes the corresponding soft predictions. $h_g(\cdot; \theta_g^{cls,t})$ is the global classifier, and $T_s$ is the server-side distillation temperature. Meanwhile, the aggregated local soft prediction for each class $c$ is computed as:

$$
\bar{\boldsymbol{p}}^c = \frac{1}{|\mathcal{S}_c^t|} \sum_{i\in\mathcal{S}_c^t} \boldsymbol{p}_i^c,
$$
(9)

where $\mathcal{S}_c^t$ denotes the set of clients that contain class $c$. The server-side mutual distillation loss is defined as:

$$
\mathcal{L}_{\text{mutual}} = \frac{1}{K} \sum_{c=1}^{K} \left[ \text{KL}\left(\boldsymbol{p}_g^c \parallel \bar{\boldsymbol{p}}^c\right) + \eta_s \cdot \text{KL}\left(\bar{\boldsymbol{p}}^c \parallel \boldsymbol{p}_g^c\right) \right],
$$
(10)

where $\eta_s$ is used to balance the two KL terms. Since $\bar{\boldsymbol{p}}^c$ is a fixed target obtained via server-side aggregation, mutual distillation does not introduce gradient conflicts and can be used to further refine the global model. The first term encourages $\boldsymbol{p}_g^c$ to align with the consensus $\bar{\boldsymbol{p}}^c$; the second term regularizes deviations, thereby stabilizing the optimization process.

### 3.3. Hierarchical Wasserstein Aggregation

Traditional prototype aggregation relies on simple averaging (Chen et al., 2023), failing to capture geometric structure under non-IID data. Although FedGMKD introduces Gaussian Mixture Models (Zhang et al., 2024), its weighting strategy depends on heuristic design. To address these limitations, we propose HWA. HWA performs geometry-aware

aggregation at both intra-class and inter-class levels. Intra-class aggregation uses Wasserstein barycenter to preserve geometric relationships. Inter-class aggregation uses Sliced Wasserstein distance to enable cross-class knowledge transfer.

### 3.3.1. INTRA-CLASS WASSERSTEIN BARYCENTER AGGREGATION

At the end of round $t$, the server collects the local prototypes $\{\boldsymbol{\mu}_i^c\}_{c \in \mathcal{C}_i}$ from participating clients. For each class $c$, the server aggregates the local prototypes in the set $\mathcal{S}_c^t$ into a global representation.

**Wasserstein Barycenter Formalization.** We model prototype aggregation as finding the Wasserstein barycenter in the space of probability measures. Grounded in optimal transport, this approach seeks a "central" distribution that minimizes transport cost to all local distributions. Treating each local prototype $\boldsymbol{\mu}_i^c$ as a Dirac measure $\delta_{\boldsymbol{\mu}_i^c}$, the global prototype $\boldsymbol{\mu}_g^c$ is obtained by solving:

$$
\begin{aligned}
\boldsymbol{\mu}_g^c &= \arg \min_{\boldsymbol{\mu} \in \mathbb{R}^d} \sum_{i \in \mathcal{S}_c^t} w_i^c \cdot W_2^2(\delta_{\boldsymbol{\mu}}, \delta_{\boldsymbol{\mu}_i^c}) \\
&= \arg \min_{\boldsymbol{\mu} \in \mathbb{R}^d} \sum_{i \in \mathcal{S}_c^t} w_i^c \cdot \|\boldsymbol{\mu} - \boldsymbol{\mu}_i^c\|_2^2,
\end{aligned}
\tag{11}
$$

where $W_2(\cdot, \cdot)$ denotes the 2-Wasserstein distance, $\delta_{\boldsymbol{x}}$ the Dirac measure centered at $\boldsymbol{x}$, $\|\cdot\|_2$ the $L_2$ norm, and $w_i^c$ the aggregation weight. The closed-form solution is the weighted average (as shown in Eq. (12)). Furthermore, the aggregated global prototype $\boldsymbol{\mu}_g^c$, global variance $\boldsymbol{\sigma}_g^c$, and global soft prediction $\boldsymbol{p}_g^c$ are:

$$
\boldsymbol{\mu}_g^c = \sum_{i \in \mathcal{S}_c^t} w_i^c \cdot \boldsymbol{\mu}_i^c,
\tag{12}
$$

$$
\boldsymbol{\sigma}_g^c = \sum_{i \in \mathcal{S}_c^t} w_i^c \cdot \boldsymbol{\sigma}_i^c,
\tag{13}
$$

$$
\boldsymbol{p}_g^c = \sum_{i \in \mathcal{S}_c^t} w_i^c \cdot \boldsymbol{p}_i^c.
\tag{14}
$$

Here, $\boldsymbol{\sigma}_i^c$ is client $i$'s local variance for class $c$.

**Distribution-Aware Adaptive Weighting.** To handle non-IID data, the adaptive weights should jointly account for data cardinality and distribution skewness. Specifically, we first compute client $i$'s normalized sample proportion for class $c$ within the participating set $\mathcal{S}_c^t$ as $n_i^c = \frac{|\mathcal{D}_i^c|}{\sum_{j \in \mathcal{S}_c^t} |\mathcal{D}_j^c|}$. Next, quantify the distribution skewness $d_i$ via the $L_2$ distance between the client's class distribution $\mathbf{Q}_i$ and the uniform distribution $\mathbf{U}$:

$$
d_i = \|\mathbf{Q}_i - \mathbf{U}\|_2,
\tag{15}
$$

where $\mathbf{Q}_i = \left[\frac{D_i^1}{D_i}, \ldots, \frac{D_i^K}{D_i}\right]$ and $\mathbf{U} = \left[\frac{1}{K}, \ldots, \frac{1}{K}\right]$, with $D_i^c$ the sample count of class $c$. The aggregation weights

are adjusted via a ReLU function:

$$
w_i^c = \frac{\text{ReLU}(n_i^c - \zeta \cdot d_i + \epsilon)}{\sum_{j \in \mathcal{S}_c^t} \text{ReLU}(n_j^c - \zeta \cdot d_j + \epsilon)},
\tag{16}
$$

where $\zeta > 0$ and $\epsilon > 0$ control the influence of distributional bias. This design ensures that clients with larger data volumes and smaller distributional bias receive higher weights.

### 3.3.2. INTER-CLASS SLICED WASSERSTEIN REFINEMENT

Aggregating only within classes may overlook semantic relationships between classes, leading to an imbalanced distribution of global prototypes in the feature space. To preserve inter-class geometric structure and realize cross-class knowledge transfer, we introduce the Sliced Wasserstein Distance (SWD) to measure relationships between class prototypes and perform inter-class refinement based on the resulting distance matrix.

**Distance Matrix Construction.** We use SWD to quantify the similarity between global prototypes $\boldsymbol{\mu}_g^c$ and $\boldsymbol{\mu}_g^{c'}$, which efficiently approximates high-dimensional Wasserstein distances via one-dimensional projections:

$$
D_{\text{SW}}[c, c'] = \left( \frac{1}{N_p} \sum_{k=1}^{N_p} W_2^2(\pi_{\theta_k} \delta_{\boldsymbol{\mu}_g^c}, \pi_{\theta_k} \delta_{\boldsymbol{\mu}_g^{c'}}) \right)^{1/2},
\tag{17}
$$

where $N_p$ is the number of projections, $\theta_k$ are random directions sampled from the unit sphere, and $\pi_{\theta_k}$ is the projection operator along direction $\theta_k$. This yields the pairwise distance matrix $\mathbf{D}_{\text{SW}}$.

**Top-$k$ Similar Class Selection.** For each class $c$, we identify its $k$ most similar classes from $\mathbf{D}_{\text{SW}}$. The set of similar classes is defined as:

$$
\begin{aligned}
\mathcal{N}_c^k &= \{c_1, \ldots, c_k\} \\
&= \arg \min_{\{c_1, \ldots, c_k\} \subset \{1, \ldots, K\} \setminus \{c\}} \sum_{j=1}^{k} D_{\text{SW}}[c, c_j],
\end{aligned}
\tag{18}
$$

where $\mathcal{N}_c^k$ contains the $k$ classes most similar to class $c$.

**Cross-Class Knowledge Transfer.** We refine the global prototypes across classes using the knowledge of similar classes, enhancing each class's prototype by fusing information from its semantic neighbors. To ensure geometric consistency across statistical moments, we apply a unified refinement strategy. The refined global prototype is computed as:

$$
\tilde{\boldsymbol{\mu}}_g^c = (1 - \lambda)\boldsymbol{\mu}_g^c + \frac{\lambda}{k} \sum_{c' \in \mathcal{N}_c^k} \boldsymbol{\mu}_g^{c'},
\tag{19}
$$

where $\lambda \in (0, 1)$ is the inter-class fusion coefficient controlling the degree of knowledge integration. This refinement realizes cross-class knowledge transfer: the prototype of class $c$ not only preserves its own representation but also absorbs complementary information from similar classes. Correspondingly, to maintain consistency in the feature distribution, the global variance $\tilde{\boldsymbol{\sigma}}_g^c$ and the global soft prediction $\tilde{\boldsymbol{p}}_g^c$ are refined using the same coefficient $\lambda$:

$$\tilde{\boldsymbol{\sigma}}_g^c = (1 - \lambda)\boldsymbol{\sigma}_g^c + \frac{\lambda}{k}\sum_{c' \in \mathcal{N}_c^k}\boldsymbol{\sigma}_g^{c'}, \tag{20}$$

$$\tilde{\boldsymbol{p}}_g^c = (1 - \lambda)\boldsymbol{p}_g^c + \frac{\lambda}{k}\sum_{c' \in \mathcal{N}_c^k}\boldsymbol{p}_g^{c'}. \tag{21}$$

Performing local updates over biased client distributions may lead to fragmentation of the learned feature manifold. Inter-class refinement can serve as a geometric regularizer, alleviating representation collapse by smoothing low-density regions, while simultaneously preserving intra-class compactness and inter-class separability.

### 3.4. Prototype-Variance Dual Alignment Mechanism

In client-side personalized distillation, clients align with global knowledge via KL divergence on the softmax output layer. However, output-space alignment alone is insufficient, as softmax projection compresses high-dimensional features into low-dimensional logits, causing information loss. To address this, we propose PVDA, which jointly matches first- and second-order moments in the feature space. Prototype alignment minimizes $L_2$ distances, while variance alignment regularizes intra-class dispersion, providing stronger regularization.

#### 3.4.1. PROTOTYPE ALIGNMENT

The local feature extractor $f_i(\cdot; \theta_i^{fe})$ is designed to produce representations consistent with the global latent space. Prototype alignment enforces first-order moment matching by minimizing the $L_2$ distance between local features and their corresponding global prototypes. The prototype alignment loss is defined as:

$$\mathcal{L}_{\text{proto}}(\mathcal{D}_i, \theta_i^{fe}) = \frac{\alpha}{|\mathcal{D}_i^{\mathcal{C}_g}|}\sum_{(\mathbf{x}, y) \in \mathcal{D}_i^{\mathcal{C}_g}}\|f_i(\mathbf{x}; \theta_i^{fe}) - \boldsymbol{\mu}_g^y\|_2^2, \tag{22}$$

where $\mathcal{C}_g$ denotes the set of classes for which global prototypes exist, and $\mathcal{D}_i^{\mathcal{C}_g}$ is the subset of client $i$'s samples belonging to these classes. $\boldsymbol{\mu}_g^y$ is the global prototype for label $y$, and $\alpha$ is a weighting hyperparameter. By minimizing $\mathcal{L}_{\text{proto}}$, the local model learns to map data into a feature space aligned with the global consensus.

#### 3.4.2. VARIANCE ALIGNMENT

Prototype alignment matches only the center of the feature distribution while ignoring its shape and dispersion. Under heterogeneous client distributions, local features may deviate from the global representation in both mean and variance. Ignoring variance may lead to feature collapse or blurred decision boundaries. To achieve finer-grained alignment, we introduce variance alignment to match the second-order moment.

For client $i$ and class $c$, let $\mathcal{F}_i^c$ denote its local feature set. The element-wise variance is defined as:

$$\boldsymbol{\sigma}_i^c = \frac{1}{|\mathcal{F}_i^c|}\sum_{\mathbf{f} \in \mathcal{F}_i^c}(\mathbf{f} - \bar{\mathbf{f}}_i^c) \odot (\mathbf{f} - \bar{\mathbf{f}}_i^c), \tag{23}$$

where $\bar{\mathbf{f}}_i^c$ is the local prototype and $\odot$ denotes the Hadamard product. The variance alignment loss is defined as:

$$\mathcal{L}_{\text{var}}(\mathcal{D}_i, \theta_i^{fe}) = \frac{\gamma}{|\mathcal{C}_i \cap \mathcal{C}_g|}\sum_{c \in \mathcal{C}_i \cap \mathcal{C}_g}\|\boldsymbol{\sigma}_i^c - \boldsymbol{\sigma}_g^c\|_2^2, \tag{24}$$

where $\gamma$ is a hyperparameter. PVDA jointly constrains the location and scale of the feature distribution: prototype alignment ensures semantic consistency, whereas variance alignment regularizes intra-class dispersion and uncertainty. Notably, variance alignment incurs negligible overhead, as it only transmits per-class element-wise variance vectors of the same dimensionality as prototypes, computed and communicated alongside prototype statistics.

### 3.5. Local Training Objective

Integrating the components introduced in Sections 3.2, 3.3, and 3.4, the complete training objective for client $i$ at round $t$ is defined as:

$$\begin{aligned}
\mathcal{L}(\mathcal{D}_i, \theta_i) = {} & \frac{1}{|\mathcal{D}_i|}\sum_{(\mathbf{x}, y) \in \mathcal{D}_i}\ell(h_i(f_i(\mathbf{x}; \theta_i^{fe}); \theta_i^{cls}), y) \\
& + \frac{\xi}{|\mathcal{C}_g|}\sum_{c \in \mathcal{C}_g}\text{KL}(\mathbf{Q}_{\text{proto}}[c, :] \,\|\, \boldsymbol{p}_g^c) \\
& + \frac{\alpha}{|\mathcal{D}_i^{\mathcal{C}_g}|}\sum_{(\mathbf{x}, y) \in \mathcal{D}_i^{\mathcal{C}_g}}\|f_i(\mathbf{x}; \theta_i^{fe}) - \boldsymbol{\mu}_g^y\|_2^2 \\
& + \frac{\gamma}{|\mathcal{C}_i \cap \mathcal{C}_g|}\sum_{c \in \mathcal{C}_i \cap \mathcal{C}_g}\|\boldsymbol{\sigma}_i^c - \boldsymbol{\sigma}_g^c\|_2^2,
\end{aligned} \tag{25}$$

where $\xi, \alpha, \gamma \geq 0$ are balancing hyperparameters. In the first training round ($t = 1$), due to the absence of global knowledge ($\mathcal{C}_g = \emptyset$), the client optimizes only the supervised loss $\mathcal{L}_{\text{CE}}$. Starting from $t \geq 2$, the full objective is enabled. This composite loss synergizes four optimization objectives: The first is the supervised learning loss, which ensures the model's discriminative capability on local private data. The second is the decoupled distillation loss,

which promotes adaptive absorption of global knowledge via client-side personalized distillation, thereby avoiding gradient conflicts. The third is the prototype alignment loss, which aligns the first-order moment of local features with the global prototype to enforce semantic consistency. Finally, the fourth is the variance alignment loss, which matches the second-order moment to regularize feature dispersion and prevent feature collapse.

### 3.6. Convergence Analysis

To establish the theoretical foundation of FedCDWA, we analyze its convergence properties under heterogeneous data distributions. The detailed assumptions and proofs are provided in Appendix G.

**Theorem 3.1** (FedCDWA Convergence). *Under Assumptions 1-5, for an arbitrary client $i$, after each communication round, the expected loss function is bounded as:*

$$
\mathbb{E}\left[\mathcal{L}_i^{\frac{1}{2},t+1}\right] \le \mathcal{L}_i^{\frac{1}{2},t} - \sum_{e=\frac{1}{2}}^{E-1}\left(\eta_e - \frac{\eta_e^2 L_1}{2}\right)\left\|\nabla\mathcal{L}^{e,t}\right\|_2^2
$$

$$
+ 2\eta_0^2 L_1 E^2 V^2 + \frac{\eta_0^2 L_1 E}{2}\sigma^2 + 4\gamma L_2\eta_0 EVM_\sigma
$$

$$
+ \xi C_{KL}L_2L_3\eta_0 EV + \xi C_{KL}M_p
$$

$$
+ 4\alpha L_2(R_f + R_\mu)\eta_0 EVM_\mu. \quad (26)
$$

**Theorem 3.2** (FedCDWA Convergence Rate). *Under Assumptions 1-5 (Appendix G), if the learning rate is set to $\eta_e = \eta_0$ for all local epochs, then after $T$ rounds ($\epsilon > 0$), it holds that*

$$
\frac{1}{TE}\sum_{t=1}^{T}\sum_{e=\frac{1}{2}}^{E-1}\left\|\nabla\mathcal{L}^{e,t}\right\|_2^2 \le \epsilon, \quad (27)
$$

*provided the total number of rounds satisfies*

$$
T \ge \frac{2\Delta}{\epsilon E(2\eta_0 - \eta_0^2 L_1) - 4\eta_0^2 L_1 E^2 V^2 - \eta_0^2 L_1 E\sigma^2 - 2C_{total}}. \quad (28)
$$

*where $\Delta = \mathcal{L}^{\frac{1}{2},1} - \mathcal{L}^*$ is the optimality gap, $\mathcal{L}^*$ is the optimal loss, and $C_{total} = \xi C_{KL}L_2L_3\eta_0 EV + \xi C_{KL}M_p + 4\alpha L_2(R_f + R_\mu)\eta_0 EVM_\mu + 4\gamma L_2\eta_0 EVM_\sigma$.*

## 4. Experiments

### 4.1. Experimental Settings.

This section evaluates FedCDWA under highly non-IID federated settings. First, we introduce the datasets, model architectures, and baselines. Then, we compare FedCDWA with competing methods in terms of overall performance. Implementation details, hyperparameter settings, and additional analyses are provided in Appendices B–F.

### 4.2. Datasets and Models.

We evaluate FedCDWA on three standard datasets: SVHN (Netzer et al., 2011), CIFAR-10, and CIFAR-100 (Krizhevsky, 2009). SVHN consists of street-view digitized images, which naturally induce non-IID scenarios due to diverse imaging conditions. CIFAR-10 and CIFAR-100 contain natural images with 10 and 100 classes, respectively. CIFAR-100 poses greater challenges for FL due to its label sparsity and fine-grained inter-class distinctions. For all tasks, we adopt ResNet-18 (He et al., 2016) as the feature extractor, adjusting the final fully connected layer to match the class dimensions.

### 4.3. Baselines.

We conduct a comprehensive comparison between FedCDWA and nine SOTA FL baselines, including FedAvg (McMahan et al., 2017), FedProx (Li et al., 2020), MOON (Li et al., 2021a), FedMD (Li & Wang, 2019), FedGen (Zhu et al., 2021), FedProto (Tan et al., 2022a), FPL (Huang et al., 2023), FedHKD (Chen et al., 2023), and FedGMKD (Zhang et al., 2024). These methods enhance the stability, personalization capability, and knowledge transfer performance of FL under non-IID settings from different perspectives, providing strong comparative baselines for assessing the benefits of FedCDWA.

### 4.4. Overall Performance on FL Methods

Table 1 summarizes the comparative results on non-IID data partitions. These partitions are generated by a Dirichlet distribution (Zhu et al., 2021) with parameter $\beta = 0.5$. We evaluate all methods in three configurations with 10, 20, and 50 clients. In these configurations, each client holds 10%, 20%, and 50% of the full dataset, respectively. The average per-round wall-clock time per client is measured in a parallel setting on a single NVIDIA RTX 3090 GPU. The local and global accuracy reported in the table are obtained by independently repeating each experiment five times with fixed random seeds. We average the final test accuracy of each round.

The proposed FedCDWA exhibits consistently strong performance across all three datasets. It generally outperforms existing FL methods in both local and global accuracy. On SVHN, compared with FedAvg, FedCDWA improves local accuracy by 1.30% to 4.22% across different numbers of clients. It also improves global accuracy by 1.69% to 3.02%. Compared with FedProto, FedCDWA improves local accuracy by 1.21% to 1.64% and global accuracy by 1.77% to 2.74%. Relative to FedGMKD, it improves local accuracy by 1.42% to 1.84% and global accuracy by 0.96% to 1.80%. FedHKD performs well overall on this dataset. Nevertheless, FedCDWA still achieves up to 2.13% higher

*Table 1.* Overall comparison of SOTA FL methods on non-IID data partitions. For each dataset and number of clients, we report local and global test accuracy (%), while the last two columns give the average per-round wall-clock time per client (s) and whether public data is used.

| Dataset | Method | Local Acc | | | Global Acc | | | Time | Pub |
|---|---|---|---|---|---|---|---|---|---|
| | | 10 | 20 | 50 | 10 | 20 | 50 | (s) | Data |
| SVHN | FedAvg | 83.34 | 86.83 | 87.02 | 83.65 | 86.32 | 89.84 | 2.49 | No |
| | FedProx | 84.39 | 87.07 | 86.81 | 84.35 | 87.89 | 89.04 | 3.46 | No |
| | Moon | 83.73 | 84.98 | 85.52 | 84.22 | 87.69 | 89.85 | 2.57 | No |
| | FedMD | 79.18 | 84.74 | 85.73 | 78.53 | 85.39 | 86.73 | 2.04 | Yes |
| | FedGen | 83.68 | 83.19 | 86.17 | 84.23 | 87.56 | 89.14 | 2.06 | No |
| | FedProto | 85.92 | 86.58 | 87.37 | 85.85 | 88.07 | 88.79 | 2.09 | No |
| | FedHKD | 85.43 | 86.46 | 87.23 | 85.49 | 87.49 | 89.17 | 2.57 | No |
| | FPL | 84.52 | 84.86 | 87.18 | 84.21 | 87.71 | 89.84 | 4.82 | No |
| | FedGMKD | 85.72 | 86.60 | 87.16 | 84.81 | 88.02 | 90.12 | 2.57 | No |
| | **FedCDWA** | **87.56** | **88.13** | **88.58** | **86.67** | **88.74** | **91.53** | 2.49 | No |
| CIFAR-10 | FedAvg | 50.75 | 59.36 | 61.31 | 49.12 | 52.47 | 56.32 | 2.67 | No |
| | FedProx | 57.80 | 63.50 | 67.49 | 48.81 | 52.03 | 57.42 | 4.64 | No |
| | Moon | 55.21 | 62.17 | 63.56 | 49.03 | 52.58 | 58.69 | 3.48 | No |
| | FedMD | 56.28 | 62.03 | 65.29 | 46.04 | 46.89 | 50.08 | 2.73 | Yes |
| | FedGen | 51.46 | 60.16 | 60.12 | 49.38 | 52.78 | 56.37 | 2.76 | No |
| | FedProto | 57.01 | 62.05 | 65.08 | 49.67 | 52.47 | 54.64 | 2.77 | No |
| | FedHKD | 54.42 | 63.53 | 65.26 | 49.73 | 51.64 | 59.31 | 3.42 | No |
| | FPL | 52.14 | 60.39 | 63.49 | 49.91 | 52.80 | 58.56 | 6.56 | No |
| | FedGMKD | 56.19 | 63.41 | 64.13 | 49.71 | 53.05 | 59.89 | 3.44 | No |
| | **FedCDWA** | **61.25** | **65.24** | **67.74** | **51.38** | **53.98** | **61.15** | 3.35 | No |
| CIFAR-100 | FedAvg | 17.50 | 19.86 | 22.40 | 15.08 | 19.62 | 24.35 | 2.67 | No |
| | FedProx | 17.16 | 22.14 | 24.29 | 14.01 | 20.60 | 25.49 | 4.65 | No |
| | Moon | 16.32 | 20.94 | 23.22 | 14.31 | 18.44 | 25.82 | 3.47 | No |
| | FedMD | 15.23 | 21.11 | 23.76 | 13.17 | 17.58 | 24.88 | 2.74 | Yes |
| | FedGen | 14.43 | 17.54 | 23.04 | 14.49 | 18.89 | 26.46 | 2.76 | No |
| | FedProto | 17.66 | 20.22 | 23.90 | 15.49 | 18.84 | 26.23 | 2.78 | No |
| | FedHKD | 17.05 | 22.03 | 23.47 | 16.89 | 20.81 | 27.35 | 3.43 | No |
| | FPL | 16.95 | 20.46 | 24.56 | 13.77 | 18.21 | 24.56 | 6.61 | No |
| | FedGMKD | 15.73 | 18.64 | 23.63 | 16.26 | 21.28 | 27.37 | 3.44 | No |
| | **FedCDWA** | **20.36** | **22.75** | **24.69** | **18.01** | **21.70** | **28.47** | 3.39 | No |

local accuracy and up to 2.36% higher global accuracy.

On CIFAR-10, compared with FedAvg, FedCDWA increases local accuracy by 5.88% to 10.50% across different numbers of clients. It also increases global accuracy by 1.51% to 4.83%. Compared with FedProto, FedCDWA improves local accuracy by 2.66% to 4.24% and global accuracy by 1.51% to 6.51%. Relative to FPL, FedCDWA consistently achieves higher local accuracy under all three client configurations, with gains of 4.25% to 9.11%. The corresponding global accuracy is also improved by 1.18% to 2.59% in all settings. FedGMKD is highly competitive on this dataset. Nevertheless, FedCDWA still surpasses it in both local and global accuracy. This indicates that the proposed method enjoys stable and substantial advantages on the non-IID CIFAR-10 partitions.

Moreover, on CIFAR-100, FedCDWA likewise achieves robust improvements over several competing methods, including FedAvg, FedProx, and FPL. For example, relative to FedAvg, it improves local accuracy by 2.29% to 2.89% and global accuracy by 2.08% to 4.12%. Compared with FedGMKD, FedCDWA improves local accuracy by 1.06%

to 4.11% and global accuracy by 1.75% to 3.84%. These advantages are particularly pronounced when the number of clients is small (corresponding to a higher degree of non-IIDness). This suggests that FedCDWA can still maintain strong global performance in more challenging, highly heterogeneous settings.

In terms of computational efficiency, FedCDWA incurs only a moderate increase in training time over FedAvg on CIFAR-10 and CIFAR-100. The per-round time on SVHN is almost identical. Compared with FedProx, FPL, and FedGMKD, FedCDWA attains higher accuracy in most settings with comparable or even lower computational cost. Relative to FedMD, FedCDWA has a similar order of training time but achieves superior local and global accuracy. Overall, FedCDWA strikes a favorable balance between accuracy and computational cost. Its additional computational overhead is well compensated by the consistent gains in both local and global accuracy.

## 5. Conclusion

In this paper, we propose FedCDWA, a novel FL framework that enables effective prototype distillation under data heterogeneity. FedCDWA addresses three critical challenges in federated prototype learning: gradient conflicts in coupled distillation, geometric distortion in prototype aggregation, and incomplete distribution alignment. Through DDS, HWA, and PVDA, FedCDWA achieves robust knowledge transfer and improved generalization across heterogeneous clients. On CIFAR-10/100 and SVHN, FedCDWA outperforms SOTA across heterogeneity levels, with up to 10.5% accuracy gains. We prove rigorous convergence and quantify the trade-off between local personalization and global consistency. HWA adds moderate overhead from Wasserstein distance computation, but the gains justify the cost. Future work will improve efficiency and extend to cross-device and dynamic-client settings.

## Impact Statement

This paper presents work whose goal is to advance FL under data heterogeneity. FedCDWA enables collaborative model training without sharing raw data, thereby enhancing privacy protection in distributed machine learning scenarios. The proposed method has potential applications in privacy-sensitive domains such as healthcare, finance, and mobile computing. These domains cannot centralize data due to privacy regulations or practical constraints.

FedCDWA addresses data heterogeneity through DDS, HWA, and PVDA. This approach contributes to more equitable federated learning systems by improving performance across diverse client populations. The framework preserves both global consistency and local personalization. This makes it particularly suitable for applications where different clients have distinct data characteristics and requirements.

There are many potential societal consequences of our work, none of which we feel must be specifically highlighted beyond the above considerations.

## Acknowledgements

This work was supported in part by the National Key R&D Program of China under Grant 2024YFB3108101; the National Cryptologic Science Fund of China under Grant 2025NCSF02038; the National Natural Science Foundation of China under Grant 62372356; the Key Laboratory of Computing Power Network and Information Security, Ministry of Education, under Grant 2023ZD018; the Key Research and Development Program of Shaanxi under Grant 2021ZDLGY06-02; the Natural Science Foundation of Shaanxi Province under Grant 2019ZDLGY1202; the Shaanxi Innovation Team Project under Grant 2018TD-007; the Xi'an Science and Technology Innovation Plan under Grant 20189168CX9JC10; and the 111 Center under Grant B16037.

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

## A. Overview of the FedCDWA framework

FedCDWA integrates Decoupled Distillation Strategy (DDS), Hierarchical Wasserstein Aggregation (HWA), and Prototype-Variance Dual Alignment (PVDA) to achieve robust knowledge transfer under non-IID settings. The framework operates iteratively, as outlined in Algorithm 1. The server initializes the global model $\theta_g^0$ and sets the global knowledge $\mathcal{K}^0 = \emptyset$, where $\mathcal{K}$ comprises prototypes, variances, and soft predictions for each class. At each global round $t$, the server samples a subset of clients $\mathcal{S}^t$ and broadcasts $(\theta_g^{t-1}, \mathcal{K}^{t-1})$ to the selected clients. Each client $i \in \mathcal{S}^t$ initializes its local model with the received global model. In the first round ($t = 1$), clients update their models using only the cross-entropy loss. From the second round onwards ($t \geq 2$), clients perform personalized prototype inference and update their models by minimizing the complete objective. This objective comprises four components: (1) supervised cross-entropy loss for classification; (2) decoupled distillation loss from global prototypes; (3) prototype alignment loss measured by Wasserstein distance; and (4) variance alignment loss for distribution matching. After local training, clients compute local knowledge $\mathcal{K}_i^t = \{\mu_i^c, \sigma_i^c, p_i^c\}$ and upload both $\theta_i^t$ and $\mathcal{K}_i^t$ to the server. The server performs hierarchical aggregation through HWA, which conducts geometry-aware prototype aggregation and inter-class refinement based on optimal transport theory. This is followed by standard model parameter aggregation and mutual distillation for knowledge refinement. This process repeats across multiple FL rounds.

## B. General Settings

All experiments are implemented and conducted in PyTorch on a single NVIDIA RTX 3090 GPU with 24 GB of memory. We use the Adam optimizer (Kingma & Ba, 2015) with an initial learning rate of 0.001, which is exponentially decayed by a factor of $\gamma_d = 0.5$ every 10 global rounds, while the remaining momentum-related hyperparameters are kept at their default values. The number of global rounds is set to 200 for SVHN and 100 for both CIFAR-10 and CIFAR-100; accordingly, in each round, each client on SVHN performs 1 local training epoch $E$, whereas on CIFAR-10 and CIFAR-100, each client performs 2 local epochs $E$. The batch size is fixed to 64, and the client participation rate is set to 1. For all datasets (SVHN and CIFAR-10/100), the dimensionality of the latent representations is fixed to 32.

## C. Hyperparameter Settings

To ensure a fair comparison, we primarily follow the configurations recommended in the original papers of the baseline methods and keep the hyperparameters as consistent as possible across different datasets, unless otherwise specified. For FedAvg (McMahan et al., 2017), we adopt a unified optimizer configuration without introducing any additional hyperparameters and use it as the basic reference for assessing the gains of other methods; in FedProx (Li et al., 2020), the proximal regularization coefficient $\mu_{\text{prox}}$ is set to 0.5. For Moon (Li et al., 2021a), the weight of the contrastive learning term $\mu_{\text{moon}}$ is set to 0.3, and the temperature parameter in the contrastive loss is set to 0.5. In FedGen (Zhu et al., 2021), the generative model adopts a Multi-Layer Perception (MLP) architecture with a hidden dimension of 512; in each global communication round, the generator is trained for 5 epochs, the ratio between the batch size of generated data and that of local training data is 0.5 (i.e., the generated batch size is 32), and the generation-related loss weights $\alpha_{\text{generative}}$ and $\beta_{\text{generative}}$ are initialized to 10 and updated after each round with a decay factor of 0.98. For FedMD (Li & Wang, 2019), the regularization hyperparameter $\lambda_{\text{md}}$ is set to 0.05, and the size of the public dataset is the same as that of the local training dataset on a single client. For FedProto (Tan et al., 2022a), the weight of the prototype regularization term $\lambda_{\text{proto}}$ is set to 0.05; for FPL (Huang et al., 2023), the regularization coefficient $\lambda_{\text{FPL}}$ is set to 0.1, and 10 local prototypes are maintained for each class. For the hyper-knowledge- and mixed-prototype-based methods FedHKD (Chen et al., 2023) and FedGMKD (Zhang et al., 2024), we adopt the configurations recommended in their original papers: in FedHKD, the loss weights $\lambda$ and $\gamma$ are set to 0.06 and 0.05, respectively, and the temperature parameter is unified to 0.6 for both FedHKD and Moon; in FedGMKD, the hyperparameters $\lambda$ and $\gamma$ are set to 0.8 and 0.2, respectively, the coefficients $a$ and $b$ in discrepancy-aware aggregation are both set to 0.2, and the distillation temperature $T$ is set to 0.7. For our proposed FedCDWA, the client-side distillation loss weight $\xi$ is set to 0.08, the server-side mutual distillation balancing coefficient $\eta$ is set to 0.5, and the distillation temperatures on the client and server are both set to $T = T_s = 0.7$. The inter-class fusion coefficient $\lambda$ is set to 0.4, while the prototype alignment weight $\alpha$ and variance alignment weight $\gamma$ are selected via grid search from combinations of $\alpha \in \{0, 0.1, 0.2, 0.3, 0.4, 0.5, 0.6\}$ and $\gamma \in \{0, 0.05, 0.1, 0.15, 0.2, 0.25, 0.3\}$. On the server, 10 distillation update steps are performed in each communication round with a learning rate of 0.015; the inter-class fusion parameter $k_s$ is set to 5 for SVHN and CIFAR-10, and 70 for CIFAR-100, and the number of random projections for the Sliced Wasserstein distance $N_p$ is set to 256.

---

**Algorithm 1** FedCDWA: Decoupled Federated Prototype Distillation with Hierarchical Wasserstein Aggregation

---

**Require:** Clients $n$; rounds $T$; local epochs $E$; server distill steps $E_s$; participation rate $\mu$; Hyperparameters $(\xi, \alpha, \gamma, \zeta, T_c, T_s, \eta_s, \lambda, k, N_p, \epsilon)$

**Ensure:** Global model $\theta_g^T$; global knowledge $\mathcal{K}^T = \{\tilde{\mu}_g^c, \tilde{\sigma}_g^c, \tilde{p}_g^c\}_{c=1}^K$

1: Initialize $\theta_g^0$ and set $\mathcal{K}^0 \leftarrow \emptyset$
2: **for** $t = 1, \ldots, T$ **do**
3:      Sample $\mathcal{S}^t$ with $|\mathcal{S}^t| = \lfloor \mu n \rfloor$ and broadcast $(\theta_g^{t-1}, \mathcal{K}^{t-1})$ to clients
4:      **// Client Update (in parallel)**
5:      **for** each $i \in \mathcal{S}^t$ **do**
6:          Client $i$ receives $(\theta_g^{t-1}, \mathcal{K}^{t-1})$ and sets $\theta_i^{t-1} \leftarrow \theta_g^{t-1}$
7:          **if** $t = 1$ **then**
8:              Client $i$ updates $\theta_i^t$ by minimizing CE loss (Eq. 2)
9:          **else**
10:              Client $i$ computes personalized inference $\mathbf{Q}_{\text{proto}}$ (Eq. 6)
11:              Client $i$ updates $\theta_i^t$ by minimizing complete objective (Eq. 25)
12:          **end if**
13:          Client $i$ computes local knowledge $\mathcal{K}_i^t = \{\mu_i^c, \sigma_i^c, p_i^c\}_{c \in \mathcal{C}_i}$ (Eq. 23, 5)
14:          Client $i$ uploads $(\theta_i^t, \mathcal{K}_i^t)$ to server
15:      **end for**
16:      **// Server Aggregation**
17:      **// Step 1: HWA intra-class prototype aggregation**
18:      **for** $c = 1, \ldots, K$ **do**
19:          Set $\mathcal{S}_c^t \leftarrow \{i \in \mathcal{S}^t : c \in \mathcal{C}_i\}$
20:          **if** $|\mathcal{S}_c^t| > 0$ **then**
21:              Server computes distribution-aware weights $\{w_i^c\}_{i \in \mathcal{S}_c^t}$ (Eq. 16)
22:              Server aggregates prototypes $(\mu_g^c, \sigma_g^c, p_g^c)$ (Eq. 12–14)
23:          **end if**
24:      **end for**
25:      **// Step 2: HWA inter-class prototype refinement**
26:      Server computes SW distance matrix $\mathbf{D}_{\text{SW}}$ (Eq. 17)
27:      **for** $c = 1, \ldots, K$ **do**
28:          **if** $\mu_g^c$ exists **then**
29:              Server selects top-$k$ neighbors $\mathcal{N}_c^k$ (Eq. 18)
30:              Server refines prototypes $(\tilde{\mu}_g^c, \tilde{\sigma}_g^c, \tilde{p}_g^c)$ (Eq. 19–21)
31:          **end if**
32:      **end for**
33:      Set $\mathcal{K}^t \leftarrow \{\tilde{\mu}_g^c, \tilde{\sigma}_g^c, \tilde{p}_g^c\}_{c=1}^K$
34:      **// Step 3: Model parameter aggregation**
35:      Server aggregates model parameters $\theta_g^t \leftarrow \sum_{i \in \mathcal{S}^t} \rho_i \theta_i^t$ (Eq. 3)
36:      **// Step 4: Server-side mutual distillation**
37:      **if** $t \geq 2$ **then**
38:          **for** $e = 1, \ldots, E_s$ **do**
39:              Server computes global soft predictions $\{p_g^c\}$ from $\{\tilde{\mu}_g^c\}$ (Eq. 8)
40:              Server computes aggregated local soft predictions $\{\bar{p}^c\}$ (Eq. 9)
41:              Server updates $\theta_g^t$ by minimizing bidirectional mutual distillation loss (Eq. 10)
42:          **end for**
43:          Server updates $\mathcal{K}^t$ with refined $\{p_g^c\}$
44:      **end if**
45: **end for**
46: **return** $\theta_g^T$ and $\mathcal{K}^T$

---

# D. Additional Experimental Results

In this section, we present additional experimental results. These results demonstrate the robustness and generalization capability of FedCDWA under various conditions. We evaluate FedCDWA under different data heterogeneity levels, model architectures, and runtime profiling settings. We compare it with SOTA baseline methods.

### D.1. Effect of Data Heterogeneity

To assess the robustness of FedCDWA to data heterogeneity, we conduct experiments on SVHN and CIFAR-10. We vary the heterogeneity level $\beta \in \{0.2, 0.8, 2.0, 5.0\}$. The degree of heterogeneity is controlled by the Dirichlet distribution parameter $\beta$. Smaller $\beta$ values concentrate each client's data on fewer classes, yielding stronger non-IID settings. Larger $\beta$ values lead to more uniform label distributions across clients, approaching IID settings.

*Table 2.* Local and global accuracy (%) of different methods on CIFAR-10 under varying heterogeneity levels $\beta$.

| Method | Local Acc | | | | Global Acc | | | |
|---|---|---|---|---|---|---|---|---|
| | $\beta = 0.2$ | $\beta = 0.8$ | $\beta = 2.0$ | $\beta = 5.0$ | $\beta = 0.2$ | $\beta = 0.8$ | $\beta = 2.0$ | $\beta = 5.0$ |
| FedAvg | 65.09 | 51.19 | 46.70 | 43.53 | 40.62 | 50.72 | 52.02 | 52.06 |
| FedProx | 70.59 | 50.39 | 47.03 | 42.45 | 42.71 | 50.35 | 52.11 | 52.50 |
| Moon | 70.26 | 51.73 | 47.65 | 42.92 | 39.28 | 50.42 | 51.30 | 52.50 |
| FedMD | 66.57 | 53.87 | 49.52 | 46.59 | 37.16 | 49.01 | 50.29 | 51.44 |
| FedGen | 65.11 | 48.85 | 45.45 | 43.81 | 40.36 | 50.40 | 51.11 | 51.48 |
| FedProto | 71.36 | 52.73 | 48.33 | 46.30 | 39.09 | 50.85 | 50.71 | 53.10 |
| FedHKD | 72.97 | 49.64 | 46.90 | 43.57 | 43.29 | 51.07 | 52.69 | 53.04 |
| FPL | 68.35 | 58.36 | 47.93 | 46.17 | 41.01 | 51.15 | 52.29 | 52.43 |
| FedGMKD | 71.58 | 50.88 | 44.66 | 43.15 | 37.47 | 51.32 | 52.40 | 53.39 |
| FedCDWA | **75.13** | **59.04** | **51.17** | **46.64** | **43.50** | **52.78** | **53.13** | **54.22** |

*Table 3.* Local and global accuracy (%) of different methods on SVHN under varying heterogeneity levels $\beta$.

| Method | Local Acc | | | | Global Acc | | | |
|---|---|---|---|---|---|---|---|---|
| | $\beta = 0.2$ | $\beta = 0.8$ | $\beta = 2.0$ | $\beta = 5.0$ | $\beta = 0.2$ | $\beta = 0.8$ | $\beta = 2.0$ | $\beta = 5.0$ |
| FedAvg | 82.86 | 82.11 | 81.71 | 80.30 | 80.37 | 84.67 | 84.40 | 85.02 |
| FedProx | 87.69 | 83.48 | 81.12 | 80.79 | 81.07 | 85.59 | 85.46 | 85.17 |
| Moon | 87.12 | 82.80 | 80.47 | 78.91 | 81.44 | 85.07 | 85.39 | 85.82 |
| FedMD | 76.68 | 81.60 | 81.46 | 79.70 | 75.19 | 83.45 | 83.65 | 83.50 |
| FedGen | 87.00 | 81.02 | 81.12 | 80.09 | 80.75 | 85.21 | 84.81 | 85.49 |
| FedProto | 86.32 | 83.69 | 82.95 | 80.85 | 80.71 | 85.36 | 85.74 | 85.70 |
| FedHKD | 86.07 | 83.81 | 82.89 | 81.01 | 81.82 | 86.02 | 85.45 | 85.93 |
| FPL | 85.14 | 81.03 | 80.85 | 80.18 | 81.16 | 84.69 | 84.84 | 85.61 |
| FedGMKD | 87.43 | 82.92 | 82.55 | 79.60 | 80.17 | 85.86 | 86.09 | 85.47 |
| FedCDWA | **88.41** | **84.31** | **83.60** | **81.87** | **82.95** | **86.34** | **86.95** | **86.56** |

On CIFAR-10, FedCDWA achieves higher local and global accuracy than all competing methods for every value of $\beta$. In particular, under the highly heterogeneous setting with $\beta = 0.2$, FedCDWA yields substantial gains over multiple baselines. Relative to FedAvg, FedHKD, and FedGMKD, the local accuracy improves by 5.55%, 2.34%, and 0.98%. The global accuracy improves by 2.58%, 1.13%, and 2.78%. As $\beta$ increases, the client data distributions become more uniform. The performance gaps among different methods narrow. Yet FedCDWA remains the best-performing approach, effectively mitigating the performance degradation caused by label distribution shift.

On SVHN, the impact of data heterogeneity is more pronounced. Yet FedCDWA still demonstrates strong robustness. Under the most heterogeneous setting ($\beta = 0.2$), FedCDWA achieves 75.13% local accuracy. This exceeds FedAvg, FedHKD, and FedGMKD by 10.04%, 2.16%, and 3.55%. Meanwhile, its global accuracy reaches 43.50%. This represents improvements of 2.88%, 0.21%, and 6.03% over FedAvg, FedHKD, and FedGMKD. FedCDWA also outperforms other representative methods such as FedProto and FPL (see Table 3). As $\beta$ increases from 0.2 to 5.0, all methods improve in both local and global accuracy. FedCDWA remains the best or near-best method across all configurations.

Overall, FedCDWA exhibits strong robustness to varying degrees of data heterogeneity on both SVHN and CIFAR-10.

Compared with baseline methods, FedCDWA maintains higher accuracy when $\beta$ is small (strongly non-IID). It also maintains high accuracy when $\beta$ is large (close to IID). The proposed HWA and PVDA mechanisms effectively alleviate client drift and fully exploit cross-client shared knowledge. This improves both local and global model performance in complex non-IID settings.

## D.2. Impact of Model Complexity

To evaluate the impact of model complexity on federated learning algorithms, we compare ResNet-18 with ResNet-50. We use the same settings as in the main experiments. Specifically, we conduct experiments on SVHN and CIFAR-10. We report local and global accuracy for each method under two heterogeneity levels: $\beta = 0.5$ and $\beta = 5.0$. The results are summarized in Table 4 (SVHN) and Table 5 (CIFAR-10).

*Table 4.* Effect of model depth on SVHN. Local and global accuracy (%) with ResNet-18 and ResNet-50 under different heterogeneity levels.

| Method | ResNet-18 | | | | ResNet-50 | | | |
| --- | --- | --- | --- | --- | --- | --- | --- | --- |
| | Local Acc | | Global Acc | | Local Acc | | Global Acc | |
| | $\beta = 0.5$ | $\beta = 5.0$ | $\beta = 0.5$ | $\beta = 5.0$ | $\beta = 0.5$ | $\beta = 5.0$ | $\beta = 0.5$ | $\beta = 5.0$ |
| FedAvg | 83.34 | 80.30 | 83.65 | 85.02 | 81.43 | 79.70 | 75.98 | 83.50 |
| FedProx | 84.39 | 80.79 | 84.35 | 85.17 | 81.42 | 79.96 | 77.17 | 82.33 |
| Moon | 83.73 | 78.91 | 84.22 | 85.82 | 81.94 | 79.91 | 78.58 | 82.60 |
| FedMD | 79.18 | 79.70 | 78.53 | 83.50 | 81.36 | 79.66 | 81.90 | 82.00 |
| FedGen | 83.68 | 80.09 | 84.23 | 85.49 | 80.56 | 80.16 | 50.81 | 64.01 |
| FedProto | 85.92 | 80.85 | 85.85 | 85.70 | 80.71 | 80.85 | 66.65 | 79.33 |
| FedHKD | 85.43 | 81.01 | 85.49 | 85.93 | 82.30 | 80.81 | 83.02 | 85.73 |
| FPL | 84.52 | 80.18 | 84.21 | 85.61 | 81.71 | 80.18 | 80.08 | 84.71 |
| FedGMKD | 85.72 | 79.60 | 84.81 | 85.47 | 81.55 | 79.60 | 76.04 | 85.47 |
| FedCDWA | **87.56** | **81.87** | **86.67** | **86.56** | **82.93** | **81.62** | **82.09** | **86.51** |

*Table 5.* Effect of model depth on CIFAR-10. Local and global accuracy (%) with ResNet-18 and ResNet-50 under different heterogeneity levels.

| Method | ResNet-18 | | | | ResNet-50 | | | |
| --- | --- | --- | --- | --- | --- | --- | --- | --- |
| | Local Acc | | Global Acc | | Local Acc | | Global Acc | |
| | $\beta = 0.5$ | $\beta = 5.0$ | $\beta = 0.5$ | $\beta = 5.0$ | $\beta = 0.5$ | $\beta = 5.0$ | $\beta = 0.5$ | $\beta = 5.0$ |
| FedAvg | 50.75 | 43.53 | 49.12 | 52.06 | 50.68 | 43.09 | 41.77 | 48.52 |
| FedProx | 57.80 | 42.45 | 48.81 | 52.50 | 52.60 | 44.34 | 43.20 | 48.82 |
| Moon | 55.21 | 42.92 | 49.03 | 52.50 | 52.73 | 42.01 | 44.79 | 48.03 |
| FedMD | 56.28 | 46.19 | 46.04 | 51.44 | 50.96 | 42.85 | 44.01 | 48.80 |
| FedGen | 51.46 | 43.81 | 49.38 | 51.48 | 50.18 | 40.24 | 43.24 | 49.05 |
| FedProto | 57.01 | 46.10 | 49.67 | 53.39 | 54.16 | 36.89 | 40.10 | 45.32 |
| FedHKD | 54.42 | 43.57 | 49.73 | 53.04 | 51.77 | 44.45 | 42.89 | 49.07 |
| FPL | 52.14 | 46.17 | 49.91 | 53.43 | 53.57 | 45.73 | 44.71 | 48.32 |
| FedGMKD | 56.19 | 43.15 | 49.71 | 53.10 | 53.20 | 44.79 | 40.94 | 48.30 |
| FedCDWA | **61.25** | **46.64** | **51.38** | **54.22** | **57.74** | **47.29** | **44.93** | **50.01** |

On SVHN, FedCDWA achieves the best overall performance across different model architectures and $\beta$ values. With ResNet-18, FedCDWA clearly outperforms other baselines in both local and global accuracy. This holds for both $\beta = 0.5$ and $\beta = 5.0$. After switching to ResNet-50, most competing methods exhibit pronounced performance degradation in the strongly heterogeneous setting ($\beta = 0.5$). This indicates they struggle to fully exploit the representational capacity of deeper networks. In contrast, FedCDWA still attains high accuracy with ResNet-50.

On CIFAR-10, FedCDWA likewise achieves superior or comparable performance to all baselines for both network architectures. For ResNet-18, FedCDWA attains local and global accuracies of 61.25% and 51.38% at $\beta = 0.5$. At $\beta = 5.0$, it attains 46.64% and 54.22%. Both results exhibit clear advantages over competing methods. After switching to ResNet-50, FedCDWA achieves local and global accuracies of 57.74% and 44.93% at $\beta = 0.5$. At $\beta = 5.0$, it achieves 47.29% and 50.01%.

The experimental results under different model complexities show that many existing methods exhibit substantial performance degradation when switching from ResNet-18 to ResNet-50. This is especially pronounced under the strongly heterogeneous setting ($\beta = 0.5$). In contrast, FedCDWA is more robust to the choice of architecture. It can still effectively exploit the representational capacity of deeper networks. The proposed HWA and DDS not only surpass existing methods in terms of accuracy. They also offer better scalability and stability with respect to model complexity.

### D.3. Analysis of Runtime Overhead

The runtime column in Table 1 provides a compact per-client efficiency summary for the main comparison. Under the parallel evaluation setting, this value mainly reflects the runtime cost associated with client-side local training in one communication round. It is useful for comparing the practical training cost of different methods, but it does not separate client-side and server-side computation. It also does not reveal which server-side stage contributes to the additional overhead of FedCDWA.

To provide a more fine-grained efficiency analysis, we further report the overall per-round runtime of all methods. We split the runtime into client-side computation, server-side computation, and total runtime. We also decompose the server-side runtime of FedCDWA into summary aggregation, inter-class SWD refinement, and mutual distillation.

*Table 6.* Overall per-round runtime comparison. We report the runtime of client-side computation, server-side computation, and the complete round in seconds.

| Dataset | Method | Client-side | Server-side | Total |
|---------|--------|-------------|-------------|-------|
| SVHN | FedAvg | 24.87 | 0.02 | 24.89 |
| | FedProx | 39.30 | 0.02 | 39.32 |
| | Moon | 29.61 | 0.51 | 30.12 |
| | FedMD | 44.17 | 0.12 | 44.29 |
| | FedGen | 24.80 | 0.43 | 25.23 |
| | FedProto | 171.05 | 0.02 | 171.07 |
| | FedHKD | 167.33 | 0.02 | 167.35 |
| | FPL | 73.82 | 0.49 | 74.31 |
| | FedGMKD | 200.79 | 0.04 | 200.83 |
| | **FedCDWA** | **56.81** | 0.10 | **56.91** |
| CIFAR-10 | FedAvg | 28.17 | 0.02 | 28.19 |
| | FedProx | 48.82 | 0.02 | 48.84 |
| | Moon | 35.96 | 0.50 | 36.46 |
| | FedMD | 49.22 | 0.11 | 49.33 |
| | FedGen | 29.17 | 0.42 | 29.59 |
| | FedProto | 124.74 | 0.02 | 124.76 |
| | FedHKD | 126.00 | 0.02 | 126.02 |
| | FPL | 84.47 | 0.49 | 84.96 |
| | FedGMKD | 149.52 | 0.05 | 149.57 |
| | **FedCDWA** | **54.72** | 0.11 | **54.83** |
| CIFAR-100 | FedAvg | 28.29 | 0.02 | 28.31 |
| | FedProx | 48.09 | 0.02 | 48.11 |
| | Moon | 35.54 | 0.51 | 36.05 |
| | FedMD | 48.51 | 0.12 | 48.63 |
| | FedGen | 29.10 | 0.42 | 29.52 |
| | FedProto | 123.88 | 0.02 | 123.90 |
| | FedHKD | 124.83 | 0.05 | 124.88 |
| | FPL | 85.44 | 0.48 | 85.92 |
| | FedGMKD | 147.86 | 0.21 | 148.07 |
| | **FedCDWA** | **55.29** | 3.25 | **58.54** |

Table 6 shows that FedCDWA maintains a reasonable overall runtime on all three datasets. Although FedCDWA introduces additional client-side objectives and server-side refinement, its total runtime remains lower than several prototype- and distillation-based baselines, including FedProto, FedHKD, FPL, and FedGMKD. This indicates that the accuracy gains of FedCDWA do not rely on prohibitive runtime cost.

Table 7 further breaks down the server-side computation of FedCDWA. On the 10-class datasets, SVHN and CIFAR-10,

*Table 7.* Breakdown of FedCDWA server-side runtime. We report the runtime of summary aggregation, inter-class SWD refinement, mutual distillation, and total server-side computation in seconds.

| Dataset | Summary aggregation | Inter-class SWD refinement | Mutual distillation | Server-side total |
|---|---|---|---|---|
| SVHN | 0.03 | 0.04 | 0.03 | 0.10 |
| CIFAR-10 | 0.03 | 0.05 | 0.03 | 0.11 |
| CIFAR-100 | 0.04 | 2.94 | 0.27 | 3.25 |

inter-class SWD refinement takes only 0.04s and 0.05s, respectively. On CIFAR-100, where the number of classes increases to 100, this component increases to 2.94s and becomes the dominant source of server-side overhead.

These results show that the additional server-side cost mainly comes from class-pairwise SWD refinement and grows with the number of classes. Nevertheless, even on CIFAR-100, the total runtime of FedCDWA remains competitive. This suggests a reasonable trade-off between model accuracy and computational cost.

# E. Ablation Studies

In this section, we conduct comprehensive ablation studies to validate the effectiveness of each component in FedCDWA. We also analyze the sensitivity to key hyperparameters. We systematically evaluate the contributions of DDS, HWA, and PVDA. We also investigate the impact of various design choices on the overall performance.

## E.1. Component Ablation

We evaluate the contributions of the key components in FedCDWA via ablation studies. Besides the full FedCDWA model, we consider three variants: FedCDWA w/o DDS, FedCDWA w/o HWA, and FedCDWA w/o PVDA. These variants remove the corresponding component while keeping all other experimental settings unchanged. We report the results on SVHN and CIFAR-10 under different data heterogeneity levels ($\beta \in \{0.5, 5.0\}$) in Table 8.

*Table 8.* Ablation on DDS, HWA, and PVDA on SVHN and CIFAR-10 under different heterogeneity levels. We report local accuracy (Local Acc) and global accuracy (Global Acc) in %.

| Method | SVHN | | | | CIFAR-10 | | | |
|---|---|---|---|---|---|---|---|---|
| | Local Acc | | Global Acc | | Local Acc | | Global Acc | |
| | $\beta = 0.5$ | $\beta = 5$ | $\beta = 0.5$ | $\beta = 5$ | $\beta = 0.5$ | $\beta = 5$ | $\beta = 0.5$ | $\beta = 5$ |
| FedAvg | 83.34 | 80.30 | 83.65 | 85.02 | 50.75 | 43.53 | 49.12 | 52.06 |
| FedCDWA w/o DDS | 85.77 | 81.56 | 85.48 | 86.31 | 57.66 | 45.65 | 50.06 | 53.45 |
| FedCDWA w/o HWA | 85.72 | 81.35 | 83.24 | 85.30 | 59.79 | 44.79 | 50.03 | 52.65 |
| FedCDWA w/o PVDA | 86.17 | 80.48 | 83.98 | 85.66 | 59.17 | 45.73 | 49.99 | 52.06 |
| FedCDWA | **87.56** | **81.87** | **86.67** | **86.56** | **61.25** | **46.64** | **51.38** | **54.22** |

Across all settings, FedCDWA achieves the best or near-best performance in both local and global accuracy. Taking the highly heterogeneous CIFAR-10 setting ($\beta = 0.5$) as an example, FedCDWA improves the local accuracy of FedAvg from 50.75% to 61.25%. It also improves the global accuracy from 49.12% to 51.38%. Disabling any single component leads to noticeable degradation. The local accuracies of the w/o DDS, w/o HWA, and w/o PVDA variants drop to 57.66%, 59.79%, and 59.17%. The corresponding global accuracies are 50.06%, 50.03%, and 49.99%. All these values are lower than those of the full FedCDWA. A similar trend is observed under weaker heterogeneity ($\beta = 5.0$). FedCDWA still outperforms all variants in both local and global metrics.

On SVHN, FedCDWA likewise achieves consistent improvements over FedAvg and all ablated variants. When $\beta = 0.5$, FedCDWA raises the local accuracy from 83.34% to 87.56%. It also raises the global accuracy from 83.65% to 86.67%. The best-performing variant, w/o PVDA, attains only 86.17% local accuracy and 83.98% global accuracy. This is still clearly inferior to the full model. Notably, under this setting, the global accuracy of w/o HWA is 83.24%. This is even slightly lower than the 83.65% of FedAvg. This further highlights the importance of HWA for building a strong global model in highly non-IID scenarios.

The three components, DDS, HWA, and PVDA, all play important and complementary roles in improving the performance of

FedCDWA. Removing any one of these modules degrades local or global accuracy across different datasets and heterogeneity levels. In contrast, the full FedCDWA that integrates all components consistently achieves the highest performance. This validates the effectiveness of the overall design.

## E.2. Effect of Alignment Weights

To investigate the influence of feature alignment weight ($\alpha$) and variance alignment weight ($\gamma$), we perform a simplified grid search on SVHN and CIFAR-10. We vary $\alpha$ and $\gamma$ to evaluate their impact on performance. The experiments involve 10 clients, each using 10% of the dataset.

*Table 9.* Effect of $\alpha$ and $\gamma$ on SVHN.

| Method | $\alpha$ | $\gamma$ | Local Acc | Global Acc |
|---|---|---|---|---|
| FedCDWA | 0.30 | 0.00 | 86.64 | 86.60 |
| | 0.30 | 0.05 | 87.49 | 87.25 |
| | 0.30 | 0.10 | 87.30 | 87.29 |
| | 0.30 | 0.15 | **87.56** | **86.67** |
| | 0.30 | 0.20 | 86.06 | 87.33 |
| | 0.30 | 0.25 | 86.85 | 87.24 |
| | 0.00 | 0.15 | 84.81 | 86.48 |
| | 0.10 | 0.15 | 85.20 | 86.97 |
| | 0.20 | 0.15 | 85.97 | 87.00 |
| | 0.40 | 0.15 | 86.56 | 86.74 |
| | 0.50 | 0.15 | 86.89 | 87.11 |
| | 0.60 | 0.15 | 87.11 | 87.20 |
| FedAvg | – | – | 83.34 | 83.65 |
| FedProto | – | – | 85.92 | 85.85 |
| FPL | – | – | 84.52 | 84.21 |
| FedGMKD | – | – | 85.72 | 84.81 |

*Table 10.* Effect of $\alpha$ and $\gamma$ on CIFAR-10.

| Method | $\alpha$ | $\gamma$ | Local Acc | Global Acc |
|---|---|---|---|---|
| FedCDWA | 0.40 | 0.00 | 59.27 | 50.21 |
| | 0.40 | 0.10 | 59.19 | 49.64 |
| | 0.40 | 0.15 | 57.11 | 51.18 |
| | 0.40 | 0.20 | **61.25** | 51.38 |
| | 0.40 | 0.25 | 56.33 | 50.74 |
| | 0.40 | 0.30 | 58.12 | 51.00 |
| | 0.00 | 0.20 | 57.58 | 50.50 |
| | 0.10 | 0.20 | 57.47 | 50.59 |
| | 0.20 | 0.20 | 56.95 | 50.66 |
| | 0.30 | 0.20 | 58.41 | 50.66 |
| | 0.50 | 0.20 | 59.56 | 50.74 |
| | 0.60 | 0.20 | 58.49 | **51.56** |
| FedAvg | – | – | 50.75 | 49.12 |
| FedProto | – | – | 57.01 | 49.67 |
| FPL | – | – | 52.14 | 49.91 |
| FedGMKD | – | – | 56.19 | 49.71 |

On SVHN (Table 9), when $\alpha = 0.3$ and $\gamma = 0.15$, the local accuracy reaches its highest value of 87.56%. The global accuracy also attains its highest value of 86.67%. Compared with FedAvg, this optimal configuration improves local accuracy by 4.22%. It also improves global accuracy by 3.02%. These improvements indicate that the PVDA mechanism of FedCDWA effectively alleviates feature drift under non-IID settings. This enhances both local personalization and global model quality.

On CIFAR-10 (Table 10), when $\alpha = 0.4$ and $\gamma = 0.2$, the local accuracy achieves its highest value of 61.25%. When $\alpha = 0.6$ and $\gamma = 0.2$, the global accuracy reaches its highest value of 51.56%. Compared with FedAvg, the configuration optimized for local accuracy improves local performance by 10.50% and global performance by 2.26%. The configuration

optimized for global accuracy improves local performance by 7.74% and global performance by 2.44%.

Across all tested configurations on both datasets, FedCDWA consistently outperforms the baseline methods FedAvg, FedProto, FPL, and FedGMKD. On SVHN, FedCDWA maintains higher accuracy than all baselines, even with suboptimal hyperparameter choices. On CIFAR-10, the best local accuracy of FedCDWA (61.25%) exceeds that of FPL (52.14%) and FedGMKD (56.19%). FedCDWA also achieves higher global accuracy than these baselines. These observations clearly demonstrate that the PVDA mechanism, by jointly constraining prototype means and feature variances, keeps the local and global models consistent in terms of first- and second-order statistics. This exhibits marked effectiveness in handling heterogeneous data distributions.

Tables 9 and 10 indicate that FedCDWA exhibits robust performance across different hyperparameter settings. However, fine-tuning $\alpha$ and $\gamma$ is crucial for maximizing performance. Therefore, striking an appropriate balance between $\alpha$ and $\gamma$ is key to further improving performance. Empirically, choosing a moderate $\alpha$ together with a relatively small $\gamma$ helps maintain stable feature distributions between local and global models. This achieves a favorable balance between personalization and generalization under non-IID scenarios.

### E.3. Effect of Distillation Epochs

To validate the effectiveness of the server-side mutual distillation mechanism in FedCDWA, we conduct ablation studies on SVHN and CIFAR-10. We vary the number of mutual distillation epochs $E_s$. We use 10 clients, each holding 10% of the data. We evaluate configurations with $E_s \in \{0, 3, 5, 8, 10, 12, 15\}$. Here, $E_s = 0$ corresponds to completely removing the mutual distillation mechanism.

*Table 11.* Effect of the number of server-side mutual distillation epochs $E_s$ on SVHN and CIFAR-10.

| Method | $E_s$ | SVHN | | CIFAR-10 | |
|---|---|---|---|---|---|
| | | Local Acc | Global Acc | Local Acc | Global Acc |
| FedAvg | – | 83.34 | 83.65 | 52.05 | 48.70 |
| FedCDWA | 3 | 87.83 | 86.02 | 56.64 | 50.28 |
| | 5 | 85.69 | 86.29 | 57.97 | 51.16 |
| | 8 | 87.56 | 86.67 | 58.96 | 50.26 |
| | 10 | 85.74 | 85.39 | 61.25 | 51.38 |
| | 12 | 85.31 | 85.26 | 59.97 | 51.32 |
| | 15 | 84.47 | 84.26 | 58.59 | 50.44 |

Table 11 shows that introducing the server-side mutual distillation mechanism leads to substantial improvements in both local and global accuracy. This is compared with the no-distillation baseline. On SVHN, using only 3 rounds of mutual distillation increases the local accuracy from 83.34% to 87.83%. It also increases the global accuracy from 83.65% to 86.02%. On CIFAR-10, using 3 rounds of distillation yields a local accuracy of 56.64%. This is 4.59% higher than FedAvg. It also increases the global accuracy from 48.70% to 50.28%, an improvement of 1.58%.

The optimal number of distillation rounds varies between datasets and evaluation metrics. On SVHN, the highest local accuracy is achieved when $E_s = 3$. The highest global accuracy occurs when $E_s = 8$. On CIFAR-10, both the highest local accuracy (61.25%) and the highest global accuracy (51.38%) are attained when $E_s = 10$. Interestingly, on CIFAR-10, performance generally improves as the number of distillation rounds increases from 3 to 10. This indicates that more complex and more strongly heterogeneous datasets benefit from more intensive distillation. However, on SVHN, excessive distillation (e.g., 15 rounds) can lead to a slight performance drop. This may be because overly strong mutual distillation constrains the capacity for local personalization.

Furthermore, the results reveal a subtle trade-off between local personalization and global generalization. For SVHN, fewer distillation rounds are more favorable for local accuracy. A moderate number of rounds better balances local and global performance. For CIFAR-10, longer distillation schedules continuously improve both metrics. This suggests that the stronger data heterogeneity in CIFAR-10 requires more iterations to effectively align local and global knowledge. On both datasets, all configurations equipped with the server-side mutual distillation mechanism substantially outperform the no-distillation baseline. This highlights the robustness of FedCDWA's decoupled distillation strategy.

### E.4. Effect of Inter-Class Fusion Parameter

We evaluate the effect of $k_s$ on CIFAR-10 and CIFAR-100 under the same settings as in the main experiments. On CIFAR-10, we test $k_s \in \{3, 5, 7, 9\}$. On CIFAR-100, we test $k_s \in \{30, 50, 70, 90\}$. The results are summarized in Table 12 and Table 13.

*Table 12.* Effect of $k_s$ on CIFAR-10.

| Method | Dataset | $\beta$ | $k_s$ | Local Acc | Global Acc |
|--------|---------|---------|-------|-----------|------------|
| FedAvg | CIFAR-10 | 0.5 | – | 50.75 | 49.12 |
| FedCDWA | CIFAR-10 | 0.5 | 3 | 59.19 | 51.12 |
| | | | 5 | **61.25** | **51.38** |
| | | | 7 | 56.69 | 50.40 |
| | | | 9 | 57.55 | 50.08 |

*Table 13.* Effect of $k_s$ on CIFAR-100.

| Method | Dataset | $\beta$ | $k_s$ | Local Acc | Global Acc |
|--------|---------|---------|-------|-----------|------------|
| FedAvg | CIFAR-100 | 0.5 | – | 17.50 | 15.08 |
| FedCDWA | CIFAR-100 | 0.5 | 30 | 18.46 | 17.39 |
| | | | 50 | 19.24 | 17.46 |
| | | | 70 | **20.36** | **18.01** |
| | | | 90 | 18.96 | 17.29 |

On CIFAR-10, compared with FedAvg, FedCDWA achieves clearly higher local and global accuracy for all values of $k_s$. This demonstrates stable robustness. The best configuration is obtained at $k_s = 5$. The local and global accuracies reach 61.25% and 51.38%. These correspond to improvements of 10.50% and 2.26% over FedAvg. When $k_s$ is too small (e.g., 3) or too large (e.g., 9), performance decreases slightly. However, it remains significantly higher than the FedAvg baseline. This is because overly small values fail to fully exploit the semantic relationships between classes. Overly large values introduce noise from weakly related classes and weaken class discriminability.

On CIFAR-100, which has more classes and finer-grained labels, both local and global accuracy improve as $k_s$ increases from 30 to 70. The best performance is achieved at $k_s = 70$. The local and global accuracies reach 20.36% and 18.01%. These represent gains of 2.86% and 2.93% over FedAvg. When $k_s$ is further increased to 90, performance exhibits a slight decline. Nevertheless, it still surpasses the FedAvg baseline. This indicates that aggregating information from too many classes leads to over-smoothed prototypes and blurred class boundaries. However, the model still maintains relatively strong performance.

Overall, FedCDWA exhibits good robustness across different choices of $k_s$. Even under suboptimal settings, it still significantly outperforms FedAvg. It achieves a favorable trade-off between performance and stability.

## F. Feature Representations

### F.1. Effect of Latent Feature Dimension

To further examine whether FedCDWA depends on a relatively low-dimensional latent representation, we increase the latent feature dimension from 32 to 64 while keeping the other settings the same as the 50-client setting in the main experiments. The results are reported in Table 14.

As shown in Table 14, FedCDWA remains competitive when the latent feature dimension is increased from 32 to 64. On SVHN, FedCDWA achieves 88.78% local accuracy and 91.72% global accuracy, outperforming all compared baselines. On CIFAR-10, FedCDWA also obtains the best local and global accuracy, reaching 66.18% and 61.43%, respectively. On the more fine-grained CIFAR-100 dataset, FedCDWA achieves 27.22% local accuracy and 30.05% global accuracy, still outperforming the strongest competing baselines.

These results indicate that FedCDWA does not rely on a single low-dimensional representation. Instead, its combination of HWA and PVDA remains effective in a higher-dimensional feature space, where preserving inter-class geometry and

*Table 14.* Performance with higher-dimensional latent features. We increase the latent feature dimension from 32 to 64 and report local and global accuracy (%) under the 50-client setting.

| Dataset | Method | Local Acc | Global Acc |
|---|---|---|---|
| SVHN | FedAvg | 84.07 | 90.37 |
| | FedProx | 86.54 | 90.52 |
| | Moon | 86.38 | 90.29 |
| | FedMD | 86.03 | 85.75 |
| | FedGen | 85.49 | 89.89 |
| | FedProto | 85.84 | 85.87 |
| | FedHKD | 88.31 | 90.75 |
| | FPL | 85.13 | 89.72 |
| | FedGMKD | 88.38 | 90.81 |
| | **FedCDWA** | **88.78** | **91.72** |
| CIFAR-10 | FedAvg | 62.77 | 58.83 |
| | FedProx | 64.87 | 59.01 |
| | Moon | 63.99 | 58.60 |
| | FedMD | 63.21 | 48.24 |
| | FedGen | 58.40 | 57.83 |
| | FedProto | 57.81 | 50.98 |
| | FedHKD | 64.23 | 60.09 |
| | FPL | 63.55 | 60.33 |
| | FedGMKD | 63.41 | 60.21 |
| | **FedCDWA** | **66.18** | **61.43** |
| CIFAR-100 | FedAvg | 24.30 | 26.70 |
| | FedProx | 25.26 | 28.54 |
| | Moon | 23.16 | 25.89 |
| | FedMD | 26.54 | 28.15 |
| | FedGen | 24.72 | 26.55 |
| | FedProto | 24.96 | 28.39 |
| | FedHKD | 26.78 | 29.09 |
| | FPL | 23.35 | 24.86 |
| | FedGMKD | 24.91 | 29.44 |
| | **FedCDWA** | **27.22** | **30.05** |

aligning fine-grained feature statistics become more important. This further supports the scalability and robustness of FedCDWA with respect to latent feature dimensionality.

### F.2. t-SNE Visualization

In this section, we conduct a comparative analysis of the quality of feature representations learned by different federated learning methods (FedAvg, FedProto, FedGMKD, and FedCDWA). We visualize projections of the global feature space on the CIFAR-10 dataset ($N = 10$, $\beta = 0.5$) using t-SNE, as shown in Fig. E.1, to evaluate each method's performance in terms of feature discriminability and geometric integrity.

As shown in Figure 2a, the feature space of FedAvg exhibits pronounced divergence, with severe inter-class overlap. This phenomenon confirms that simple parameter averaging struggles to effectively align the feature spaces of heterogeneous clients, leading to blurred decision boundaries and weak overall discriminability.

Figure 2b shows the results of FedProto. Although prototype learning improves local consistency and yields several discernible clusters, pronounced feature mixing persists in the central region. This is because naive prototype averaging relies solely on first-order statistics (the mean) while ignoring feature variance and distribution skewness, thereby limiting the model's ability to establish strict inter-class boundaries.

Examining Figure 2c, FedGMKD exhibits relatively clear decision boundaries, indicating that its class-level knowledge fusion mechanism is effective for domain alignment. However, despite achieving basic inter-class separation, its within-cluster distributions remain relatively diffuse, suggesting insufficient constraints on the internal geometric structure of the feature space.

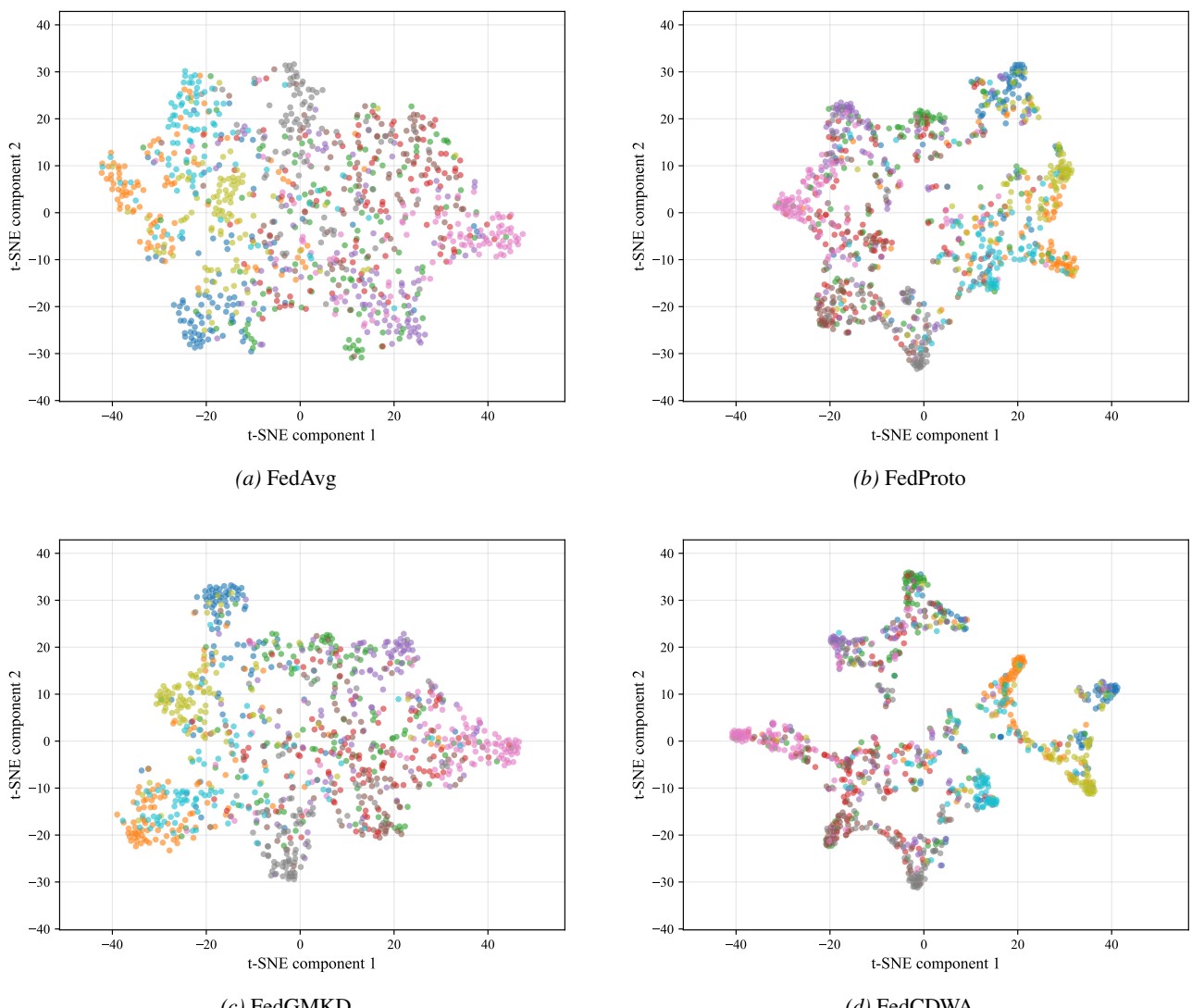

*Figure 2.* t-SNE visualization of feature representations after global aggregation on CIFAR-10 ($\beta = 0.5$). Each point represents a test sample colored by its CIFAR-10 class label. FedCDWA achieves the most compact intra-class clusters and clearest inter-class boundaries compared to baseline methods.

FedCDWA, as shown in Figure 2d, exhibits optimal feature representation quality, constructing a distinctly structured and highly discriminative feature space. First, the decoupled distillation strategy effectively circumvents gradient conflicts, establishing the feature space's global stability and purity. Building on this, HWA optimizes the geometric structure of features, enforcing sharp inter-class decision boundaries. Finally, by explicitly aligning second-order statistics, PVDA achieves exceptional intra-class compactness, effectively overcoming the common issue of diffuse feature distributions in existing methods.

## G. Convergence Analysis of FedCDWA

Let $\phi_i = (\theta_i^{fe}, \theta_i^{cls}) \in \mathbb{R}^{d_\phi}$ denote the model parameters of client $i$, where $\theta_i^{fe}$ represents the feature extractor parameters and $\theta_i^{cls}$ represents the classifier parameters. The global knowledge $\mathcal{K} = \{\boldsymbol{\mu}_g^c, \boldsymbol{\sigma}_g^c, \boldsymbol{p}_g^c\}_{c=1}^K$ consists of global prototypes, variances, and soft predictions for each class $c$.

The local objective function of client $i$ is defined as:

$$\mathcal{L}(\mathcal{D}_i, \phi_i, \mathcal{K}) = \underbrace{\frac{1}{|\mathcal{D}_i|} \sum_{(\mathbf{x},y)\in\mathcal{D}_i} \ell(h_i(f_i(\mathbf{x}; \theta_i^{fe}); \theta_i^{cls}), y)}_{\mathcal{L}_{\text{CE}}}$$

$$+ \underbrace{\frac{\xi}{|\mathcal{C}_g|} \sum_{c\in\mathcal{C}_g} \text{KL}\left(\text{softmax}\left(\frac{h_i(\boldsymbol{\mu}_g^c; \theta_i^{cls})}{T}\right) \middle\| \boldsymbol{p}_g^c\right)}_{\mathcal{L}_{\text{KD}}}$$

$$+ \underbrace{\frac{\alpha}{|\mathcal{D}_i^{\mathcal{C}_g}|} \sum_{(\mathbf{x},y)\in\mathcal{D}_i^{\mathcal{C}_g}} \|f_i(\mathbf{x}; \theta_i^{fe}) - \boldsymbol{\mu}_g^y\|_2^2}_{\mathcal{L}_{\text{proto}}}$$

$$+ \underbrace{\frac{\gamma}{|\mathcal{C}_i \cap \mathcal{C}_g|} \sum_{c\in\mathcal{C}_i\cap\mathcal{C}_g} \|\boldsymbol{\sigma}_i^c - \boldsymbol{\sigma}_g^c\|_2^2}_{\mathcal{L}_{\text{var}}}, \tag{29}$$

where the notation is consistent with Methodology Section 3.

Let $t$ denote the current global communication round. During each global round, there are $E$ local training epochs. We use the following notation:

- $e \in \{\frac{1}{2}, 1, \ldots, E\}$: local epoch index, where $e = \frac{1}{2}$ indicates the epoch immediately after receiving the aggregated global model

- $\phi_i^{E,t}$: local model of client $i$ after $E$ local epochs at round $t$

- $\phi_g^{\frac{1}{2},t+1}$: aggregated global model at round $t+1$ (before local training)

- $\mathcal{K}^t$: global knowledge at round $t$

At global round $t+1$, client $i$ initializes with the aggregated global model $\phi_g^{\frac{1}{2},t+1}$ and updated global knowledge $\mathcal{K}^{t+1}$. For convergence analysis, we use uniform aggregation weights $w_i = \frac{|D_i|}{\sum_{j=1}^m |D_j|}$ with $\sum_{i=1}^m w_i = 1$. In practice, the algorithm may use class-specific weights as described in Methodology Section 3.

The global model is updated as:

$$\phi_g^{\frac{1}{2},t+1} = \sum_{i=1}^m w_i \phi_i^{E,t}, \tag{30}$$

where $\phi_i^{E,t}$ is the local model after $E$ epochs at round $t$. The global prototypes $\boldsymbol{\mu}_g^c$, variances $\boldsymbol{\sigma}_g^c$, and soft predictions $\boldsymbol{p}_g^c$ are aggregated as:

$$\boldsymbol{\mu}_g^{c,t+1} = \sum_{i=1}^m w_i \boldsymbol{\mu}_i^{c,t}, \tag{31}$$

$$\boldsymbol{\sigma}_g^{c,t+1} = \sum_{i=1}^m w_i \boldsymbol{\sigma}_i^{c,t}, \tag{32}$$

$$\boldsymbol{p}_g^{c,t+1} = \sum_{i=1}^m w_i \boldsymbol{p}_i^{c,t}. \tag{33}$$

### G.1. Assumptions

**Assumption G.1** (Lipschitz Continuity). The gradient of the local loss function $\mathcal{L}(\cdot)$ is $L_1$-Lipschitz continuous, the embedding functions of the local feature extractor $f_i(\cdot)$ is $L_2$-Lipschitz continuous, and the embedding functions of the

local classifier $h_i(\cdot)$ is $L_3$-Lipschitz continuous:

$$\|\nabla\mathcal{L}(\phi^{t_1}) - \nabla\mathcal{L}(\phi^{t_2})\|_2 \leq L_1\|\phi^{t_1} - \phi^{t_2}\|_2, \quad \forall t_1, t_2 > 0, \tag{34}$$

$$\|f_i(\mathbf{x}; \theta_i^{fe,t_1}) - f_i(\mathbf{x}; \theta_i^{fe,t_2})\|_2 \leq L_2\|\theta_i^{fe,t_1} - \theta_i^{fe,t_2}\|_2, \quad \forall t_1, t_2 > 0, \tag{35}$$

$$\|h_i(\mathbf{f}; \theta_i^{cls,t_1}) - h_i(\mathbf{f}; \theta_i^{cls,t_2})\|_2 \leq L_3\|\theta_i^{cls,t_1} - \theta_i^{cls,t_2}\|_2, \quad \forall t_1, t_2 > 0. \tag{36}$$

Inequality (34) also implies

$$\mathcal{L}(\phi^{t_1}) - \mathcal{L}(\phi^{t_2}) \leq \langle \nabla\mathcal{L}(\phi^{t_2}), \phi^{t_1} - \phi^{t_2}\rangle + \frac{L_1}{2}\|\phi^{t_1} - \phi^{t_2}\|_2^2, \quad \forall t_1, t_2 > 0. \tag{37}$$

**Assumption G.2** (Unbiased Gradient and Bounded Variance)**.** The stochastic gradients on a batch of client $i$'s data $\xi_i$, denoted by $\mathbf{g}_i = \nabla\mathcal{L}(\phi_i, \xi_i)$, is an unbiased estimator of the local gradient:

$$\mathbb{E}_{\xi_i \sim \mathcal{D}_i}[\mathbf{g}_i] = \nabla\mathcal{L}(\phi_i), \quad \forall i \in \{1, 2, \ldots, m\}, \tag{38}$$

With the variance bounded by $\sigma^2$:

$$\mathbb{E}\left[\|\mathbf{g}_i - \nabla\mathcal{L}(\phi_i)\|_2^2\right] \leq \sigma^2, \quad \forall i \in \{1, 2, \ldots, m\}. \tag{39}$$

**Assumption G.3** (Bounded Expectation of Gradients)**.** The expectation of the stochastic gradient is bounded:

$$\mathbb{E}\left[\|\mathbf{g}_i\|_2^2\right] \leq V^2, \quad \forall i \in \{1, 2, \ldots, m\}. \tag{40}$$

**Assumption G.4** (Bounded Global Knowledge Discrepancy)**.** The discrepancy between local and global knowledge is bounded. For any client $i$ and class $c$:

$$\|\boldsymbol{\mu}_i^c - \boldsymbol{\mu}_g^c\|_2 \leq M_\mu, \tag{41}$$

$$\|\boldsymbol{\sigma}_i^c - \boldsymbol{\sigma}_g^c\|_2 \leq M_\sigma, \tag{42}$$

$$\|\boldsymbol{p}_i^c - \boldsymbol{p}_g^c\|_2 \leq M_p. \tag{43}$$

**Assumption G.5** (Bounded Feature Representations and Prototypes)**.** The feature representations and prototypes are bounded:

$$\|f_i(\mathbf{x}; \theta_i^{fe})\|_2 \leq R_f, \quad \forall \mathbf{x} \in \mathcal{D}_i, \tag{44}$$

$$\|\boldsymbol{\mu}_g^c\|_2 \leq R_\mu, \quad \forall c \in \{1, \ldots, K\}, \tag{45}$$

where $R_f, R_\mu > 0$ are constants.

### G.2. Lemmas

**Lemma G.6** (Lipschitz Continuity of KL Divergence)**.** *For probability distributions $p_1, p_2, q_1, q_2$ generated by softmax with temperature $T > 0$, there exists a constant $C_{KL} = C_{KL}(K, T)$ such that:*

$$|KL(p_1\|q_1) - KL(p_2\|q_2)| \leq C_{KL}(\|p_1 - p_2\|_2 + \|q_1 - q_2\|_2). \tag{46}$$

*Proof.* We bound the variation of the KL divergence by considering each argument separately and then combining the results.

To bound the variation with respect to $p$, we fix $q$ and consider $f(p) = \text{KL}(p\|q) = \sum_{i=1}^K p_i \log(p_i/q_i)$. The gradient with respect to $p$ is:

$$\nabla_p \text{KL}(p\|q) = \left(\log \frac{p_1}{q_1} + 1, \ldots, \log \frac{p_K}{q_K} + 1\right). \tag{47}$$

For softmax distributions with temperature $T$, each probability $p_i$ and $q_i$ satisfies $p_i, q_i \geq \epsilon > 0$ for some $\epsilon$ depending on $T$ and $K$. Specifically, for logits $z \in \mathbb{R}^K$ with bounded range, we have:

$$p_i = \frac{\exp(z_i/T)}{\sum_j \exp(z_j/T)} \geq \frac{1}{K}\exp\left(\frac{z_{\min} - z_{\max}}{T}\right) \geq \epsilon. \tag{48}$$

This lower bound ensures that $|\log(p_i/q_i)| \leq M$ for some constant $M = M(K, T, \epsilon)$. Therefore:

$$\|\nabla_p \mathrm{KL}(p\|q)\|_2 \leq \sqrt{K}(M+1), \tag{49}$$

Denoting $L_p = \sqrt{K}(M+1)$ and applying the Mean Value Theorem, there exists $p^* \in [p_1, p_2]$ such that:

$$|\mathrm{KL}(p_1\|q) - \mathrm{KL}(p_2\|q)| = |\langle \nabla_p \mathrm{KL}(p^*\|q), p_1 - p_2\rangle| \leq L_p\|p_1 - p_2\|_2. \tag{50}$$

Similarly, to bound the variation with respect to $q$, we fix $p$ and consider $g(q) = \mathrm{KL}(p\|q) = -\sum_{i=1}^{K} p_i \log q_i + C$, where $C = \sum_{i=1}^{K} p_i \log p_i$ is constant with respect to $q$. The gradient is:

$$\nabla_q \mathrm{KL}(p\|q) = \left(-\frac{p_1}{q_1}, \ldots, -\frac{p_K}{q_K}\right). \tag{51}$$

Since $p_i \leq 1$ and $q_i \geq \epsilon$, we have:

$$\|\nabla_q \mathrm{KL}(p\|q)\|_2 \leq \sqrt{\sum_{i=1}^{K} \frac{p_i^2}{q_i^2}} \leq \sqrt{\frac{K}{\epsilon^2}} = \frac{\sqrt{K}}{\epsilon}. \tag{52}$$

Denoting $L_q = \sqrt{K}/\epsilon$ and applying the Mean Value Theorem:

$$|\mathrm{KL}(p\|q_1) - \mathrm{KL}(p\|q_2)| \leq L_q\|q_1 - q_2\|_2. \tag{53}$$

To combine these bounds, we decompose the KL divergence for arbitrary $p_1, p_2, q_1, q_2$ as:

$$\mathrm{KL}(p_1\|q_1) - \mathrm{KL}(p_2\|q_2) = [\mathrm{KL}(p_1\|q_1) - \mathrm{KL}(p_1\|q_2)] + [\mathrm{KL}(p_1\|q_2) - \mathrm{KL}(p_2\|q_2)]. \tag{54}$$

Taking absolute values and using the bounds derived above:

$$|\mathrm{KL}(p_1\|q_1) - \mathrm{KL}(p_2\|q_2)| \leq |\mathrm{KL}(p_1\|q_1) - \mathrm{KL}(p_1\|q_2)| + |\mathrm{KL}(p_1\|q_2) - \mathrm{KL}(p_2\|q_2)| \tag{55}$$

$$\leq L_q\|q_1 - q_2\|_2 + L_p\|p_1 - p_2\|_2 \tag{56}$$

$$\leq C_{\mathrm{KL}}(\|p_1 - p_2\|_2 + \|q_1 - q_2\|_2), \tag{57}$$

where $C_{\mathrm{KL}} = \max\{L_p, L_q\}$. $\qquad \square$

**Lemma G.7** (Local Training Bound). *Under Assumptions G.1–G.3, the loss function after $E$ local training epochs at global round $t+1$ can be bounded as:*

$$\mathbb{E}\left[\mathcal{L}^{E,t+1}\right] \leq \mathcal{L}^{\frac{1}{2},t+1} - \sum_{e=\frac{1}{2}}^{E-1}\left(\eta_e - \frac{\eta_e^2 L_1}{2}\right)\|\nabla\mathcal{L}^{e,t+1}\|_2^2 + \frac{\eta_0^2 L_1 E}{2}\sigma^2, \tag{58}$$

*where $\eta_e$ is the learning rate at local epoch $e$.*

*Proof.* Following the standard analysis of stochastic gradient descent with $L_1$-smooth loss functions, for any local epoch $e \in \{\frac{1}{2}, 1, \ldots, E-1\}$:

$$\mathcal{L}^{e+1,t+1} \overset{(1)}{\leq} \mathcal{L}^{e,t+1} + \langle \nabla\mathcal{L}^{e,t+1}, \phi_i^{e+1,t+1} - \phi_i^{e,t+1}\rangle + \frac{L_1}{2}\|\phi_i^{e+1,t+1} - \phi_i^{e,t+1}\|_2^2$$

$$= \mathcal{L}^{e,t+1} - \eta_e\langle \nabla\mathcal{L}^{e,t+1}, \mathbf{g}^{e,t+1}\rangle + \frac{L_1}{2}\eta_e^2\|\mathbf{g}^{e,t+1}\|_2^2, \tag{59}$$

where (1) follows from Assumption G.1.

Taking expectation over the sampling batch $\xi^{e+1}$:

$$\mathbb{E}\left[\mathcal{L}^{e+1,t+1}\right] \overset{(2)}{\leq} \mathcal{L}^{e,t+1} - \eta_e\|\nabla\mathcal{L}^{e,t+1}\|_2^2 + \frac{L_1}{2}\eta_e^2\mathbb{E}\left[\|\mathbf{g}^{e,t+1}\|_2^2\right]$$

$$\overset{(3)}{=} \mathcal{L}^{e,t+1} - \eta_e\|\nabla\mathcal{L}^{e,t+1}\|_2^2 + \frac{L_1}{2}\eta_e^2\left(\|\nabla\mathcal{L}^{e,t+1}\|_2^2 + \mathbb{E}\left[\|\mathbf{g}^{e,t+1} - \nabla\mathcal{L}^{e,t+1}\|_2^2\right]\right)$$

$$\overset{(4)}{\leq} \mathcal{L}^{e,t+1} - \left(\eta_e - \frac{\eta_e^2 L_1}{2}\right)\|\nabla\mathcal{L}^{e,t+1}\|_2^2 + \frac{L_1}{2}\eta_e^2\sigma^2, \tag{60}$$

where (2) follows from Assumption G.2 (unbiasedness: $\mathbb{E}[\mathbf{g}^{e,t+1}] = \nabla\mathcal{L}^{e,t+1}$); (3) uses the variance decomposition $\mathbb{E}[\|X\|^2] = \|\mathbb{E}[X]\|^2 + \mathbb{E}[\|X - \mathbb{E}[X]\|^2]$; (4) follows from Assumption G.2 (Equation (39)).

Telescoping over all local epochs and using $\eta_{\frac{1}{2}} = \eta_0$ (non-increasing learning rate):

$$\mathbb{E}\left[\mathcal{L}^{E,t+1}\right] \leq \mathcal{L}^{\frac{1}{2},t+1} - \sum_{e=\frac{1}{2}}^{E-1}\left(\eta_e - \frac{\eta_e^2 L_1}{2}\right)\|\nabla\mathcal{L}^{e,t+1}\|_2^2 + \frac{\eta_0^2 L_1 E}{2}\sigma^2. \tag{61}$$

$$\square$$

**Lemma G.8** (Global Aggregation Bound). *Under Assumptions G.1–G.5, following the model and knowledge aggregation at the server, the loss function of any client $i$ at global round $t+1$ can be bounded as:*

$$\mathbb{E}\left[\mathcal{L}_i^{\frac{1}{2},(t+1)}\right] \leq \mathcal{L}_i^{E,t} + 2\eta_0^2 L_1 E^2 V^2 + C_{KD} + C_{proto} + C_{var}, \tag{62}$$

*where*

$$C_{KD} = \xi C_{KL} L_2 L_3 \eta_0 EV + \xi C_{KL} M_p, \tag{63}$$

$$C_{proto} = 4\alpha L_2(R_f + R_\mu)\eta_0 EV M_\mu, \tag{64}$$

$$C_{var} = 4\gamma L_2 \eta_0 EV M_\sigma. \tag{65}$$

*Proof.*

$$\mathcal{L}_i^{\frac{1}{2},(t+1)} - \mathcal{L}_i^{E,t}$$

$$= \mathcal{L}(\mathcal{D}_i, \phi_g^{\frac{1}{2},t+1}, \mathcal{K}^{t+1}) - \mathcal{L}(\mathcal{D}_i, \phi_i^{E,t}, \mathcal{K}^t)$$

$$= \underbrace{\mathcal{L}(\mathcal{D}_i, \phi_g^{\frac{1}{2},t+1}, \mathcal{K}^{t+1}) - \mathcal{L}(\mathcal{D}_i, \phi_i^{E,t}, \mathcal{K}^{t+1})}_{\text{Model update effect}}$$

$$+ \underbrace{\mathcal{L}(\mathcal{D}_i, \phi_i^{E,t}, \mathcal{K}^{t+1}) - \mathcal{L}(\mathcal{D}_i, \phi_i^{E,t}, \mathcal{K}^t)}_{\text{Knowledge update effect}}. \tag{66}$$

**Part 1: Model Update Effect.**

Using Assumption G.1:

$$\mathcal{L}(\mathcal{D}_i, \phi_g^{\frac{1}{2},t+1}, \mathcal{K}^{t+1}) - \mathcal{L}(\mathcal{D}_i, \phi_i^{E,t}, \mathcal{K}^{t+1})$$

$$\overset{(1)}{\leq} \left\langle \nabla_\phi\mathcal{L}(\mathcal{D}_i, \phi_i^{E,t}, \mathcal{K}^{t+1}), \phi_g^{\frac{1}{2},t+1} - \phi_i^{E,t}\right\rangle + \frac{L_1}{2}\|\phi_g^{\frac{1}{2},t+1} - \phi_i^{E,t}\|_2^2$$

$$\overset{(2)}{=} \left\langle \nabla_\phi\mathcal{L}^{E,t}, \sum_{j=1}^m w_j(\phi_j^{E,t} - \phi_i^{E,t})\right\rangle + \frac{L_1}{2}\left\|\sum_{j=1}^m w_j(\phi_j^{E,t} - \phi_i^{E,t})\right\|_2^2, \tag{67}$$

where (1) follows from $L_1$-smoothness (Assumption G.1); (2) follows from the aggregation rule (Equation (30)) and the fact that $\sum_{j=1}^m w_j = 1$.

Taking expectation:

$$
\mathbb{E}\left[\mathcal{L}(\mathcal{D}_i, \phi_g^{\frac{1}{2},t+1}, \mathcal{K}^{t+1}) - \mathcal{L}(\mathcal{D}_i, \phi_i^{E,t}, \mathcal{K}^{t+1})\right]
$$

$$
\overset{(3)}{\leq} \frac{L_1}{2} \mathbb{E}\left[\left\|\sum_{j=1}^{m} w_j(\phi_j^{E,t} - \phi_i^{E,t})\right\|_2^2\right]
$$

$$
\overset{(4)}{\leq} \frac{L_1}{2} \sum_{j=1}^{m} w_j \mathbb{E}\left[\|\phi_j^{E,t} - \phi_i^{E,t}\|_2^2\right]
$$

$$
\overset{(5)}{\leq} L_1 \sum_{j=1}^{m} w_j \left(\mathbb{E}\left[\|\phi_j^{E,t} - \phi_g^{\frac{1}{2},t}\|_2^2\right] + \mathbb{E}\left[\|\phi_i^{E,t} - \phi_g^{\frac{1}{2},t}\|_2^2\right]\right)
$$

$$
\overset{(6)}{\leq} 2L_1 \mathbb{E}\left[\|\phi_i^{E,t} - \phi_i^{\frac{1}{2},t}\|_2^2\right]
$$

$$
\overset{(7)}{\leq} 2L_1 E \sum_{e=\frac{1}{2}}^{E-1} \mathbb{E}\left[\|\phi_i^{e+1,t} - \phi_i^{e,t}\|_2^2\right]
$$

$$
= 2L_1 E \sum_{e=\frac{1}{2}}^{E-1} \eta_e^2 \mathbb{E}\left[\|\mathbf{g}_i^{e,t}\|_2^2\right]
$$

$$
\overset{(8)}{\leq} 2L_1 E^2 \eta_0^2 V^2, \tag{68}
$$

where (3) follows from Young's inequality ($2\langle a, b \rangle \leq \|a\|^2 + \|b\|^2$); (4) follows from Jensen's inequality; (5) follows from $\|a - b\|^2 \leq 2(\|a - c\|^2 + \|b - c\|^2)$; (6) follows from $\sum_{j=1}^{m} w_j = 1$; (7) follows from Cauchy-Schwarz inequality; (8) follows from Assumption G.3.

### Part 2: Knowledge Update Effect.

From the decomposition (66), the knowledge update effect can be written as:

$$
\mathcal{L}(\mathcal{D}_i, \phi_i^{E,t}, \mathcal{K}^{t+1}) - \mathcal{L}(\mathcal{D}_i, \phi_i^{E,t}, \mathcal{K}^t)
$$

$$
= \underbrace{\frac{\xi}{|\mathcal{C}_g|} \sum_{c \in \mathcal{C}_g} \left[\mathrm{KL}\left(Q(h_i(\boldsymbol{\mu}_g^{c,t+1}), T) \,\middle\|\, \boldsymbol{p}_g^{c,t+1}\right) - \mathrm{KL}\left(Q(h_i(\boldsymbol{\mu}_g^{c,t}), T) \,\middle\|\, \boldsymbol{p}_g^{c,t}\right)\right]}_{\Delta\mathcal{L}_{\mathrm{KD}}}
$$

$$
+ \underbrace{\frac{\alpha}{|\mathcal{D}_i^{\mathcal{C}_g}|} \sum_{(\mathbf{x},y) \in \mathcal{D}_i^{\mathcal{C}_g}} \left[\|f_i(\mathbf{x}) - \boldsymbol{\mu}_g^{y,t+1}\|_2^2 - \|f_i(\mathbf{x}) - \boldsymbol{\mu}_g^{y,t}\|_2^2\right]}_{\Delta\mathcal{L}_{\mathrm{proto}}}
$$

$$
+ \underbrace{\frac{\gamma}{|\mathcal{C}_i \cap \mathcal{C}_g|} \sum_{c \in \mathcal{C}_i \cap \mathcal{C}_g} \left[\|\boldsymbol{\sigma}_i^c - \boldsymbol{\sigma}_g^{c,t+1}\|_2^2 - \|\boldsymbol{\sigma}_i^c - \boldsymbol{\sigma}_g^{c,t}\|_2^2\right]}_{\Delta\mathcal{L}_{\mathrm{var}}}, \tag{69}
$$

where the CE term does not depend on global knowledge.

*Term 1: KD loss.* The change in KD loss due to knowledge update is:

$$
\Delta\mathcal{L}_{\mathrm{KD}} = \frac{\xi}{|\mathcal{C}_g|} \sum_{c \in \mathcal{C}_g} \left[\mathrm{KL}\left(Q(h_i(\boldsymbol{\mu}_g^{c,t+1}; \theta_i^{cls,E,t}), T) \,\middle\|\, \boldsymbol{p}_g^{c,t+1}\right)\right.
$$

$$
\left. - \mathrm{KL}\left(Q(h_i(\boldsymbol{\mu}_g^{c,t}; \theta_i^{cls,E,t}), T) \,\middle\|\, \boldsymbol{p}_g^{c,t}\right)\right], \tag{70}
$$

where $Q(\cdot, T) = \text{softmax}(\cdot/T)$. Taking absolute value and applying Lemma G.6:

$$
\begin{aligned}
|\Delta\mathcal{L}_{\text{KD}}| &\overset{(1)}{\leq} \frac{\xi C_{\text{KL}}}{|\mathcal{C}_g|} \sum_{c \in \mathcal{C}_g} \Big[ \|Q(h_i(\boldsymbol{\mu}_g^{c,t+1}), T) - Q(h_i(\boldsymbol{\mu}_g^{c,t}), T)\|_2 + \|\boldsymbol{p}_g^{c,t+1} - \boldsymbol{p}_g^{c,t}\|_2 \Big] \\
&\overset{(2)}{\leq} \frac{\xi C_{\text{KL}}}{|\mathcal{C}_g|} \sum_{c \in \mathcal{C}_g} \Big[ L_3 \|\boldsymbol{\mu}_g^{c,t+1} - \boldsymbol{\mu}_g^{c,t}\|_2 + M_p \Big] \\
&\overset{(3)}{\leq} \xi C_{\text{KL}} L_3 \max_{c \in \mathcal{C}_g} \|\boldsymbol{\mu}_g^{c,t+1} - \boldsymbol{\mu}_g^{c,t}\|_2 + \xi C_{\text{KL}} M_p \\
&\overset{(4)}{\leq} \xi C_{\text{KL}} L_2 L_3 \sum_{j=1}^{m} w_j \mathbb{E}\Big[ \|\phi_j^{E,t} - \phi_j^{\frac{1}{2},t}\|_2 \Big] + \xi C_{\text{KL}} M_p \\
&\overset{(5)}{\leq} \xi C_{\text{KL}} L_2 L_3 \eta_0 EV + \xi C_{\text{KL}} M_p,
\end{aligned}
\tag{71}
$$

where (1) follows from Lemma G.6; (2) follows from the Lipschitz property of classifier $h_i$ and Assumption G.4; (3) follows from taking the maximum over classes; (4) follows from the aggregation rule and Assumption G.1; (5) follows from $\mathbb{E}[\|\phi_j^{E,t} - \phi_j^{\frac{1}{2},t}\|] \leq \eta_0 EV$.

*Term 2: Prototype alignment loss.*

The change in prototype alignment loss is:

$$
\Delta\mathcal{L}_{\text{proto}} = \frac{\alpha}{|\mathcal{D}_i^{\mathcal{C}_g}|} \sum_{(\mathbf{x},y) \in \mathcal{D}_i^{\mathcal{C}_g}} \Big[ \|f_i(\mathbf{x}; \theta_i^{fe,E,t}) - \boldsymbol{\mu}_g^{y,t+1}\|_2^2 - \|f_i(\mathbf{x}; \theta_i^{fe,E,t}) - \boldsymbol{\mu}_g^{y,t}\|_2^2 \Big].
\tag{72}
$$

$$
\Delta\mathcal{L}_{\text{proto}} \overset{(1)}{=} \frac{\alpha}{|\mathcal{D}_i^{\mathcal{C}_g}|} \sum_{(\mathbf{x},y) \in \mathcal{D}_i^{\mathcal{C}_g}} \Big[ \|\boldsymbol{\mu}_g^{y,t+1}\|_2^2 - \|\boldsymbol{\mu}_g^{y,t}\|_2^2 - 2\langle f_i(\mathbf{x}), \boldsymbol{\mu}_g^{y,t+1} - \boldsymbol{\mu}_g^{y,t} \rangle \Big],
\tag{73}
$$

where (1) follows from the identity $\|a - b\|^2 - \|a - c\|^2 = \|b\|^2 - \|c\|^2 - 2\langle a, b - c \rangle$.

Taking absolute value and bounding each sample's contribution:

$$
\begin{aligned}
|\Delta\mathcal{L}_{\text{proto}}| &\overset{(2)}{\leq} \frac{\alpha}{|\mathcal{D}_i^{\mathcal{C}_g}|} \sum_{(\mathbf{x},y) \in \mathcal{D}_i^{\mathcal{C}_g}} \Big[ \|\boldsymbol{\mu}_g^{y,t+1} - \boldsymbol{\mu}_g^{y,t}\|_2 \\
&\qquad\qquad \cdot (\|\boldsymbol{\mu}_g^{y,t+1}\|_2 + \|\boldsymbol{\mu}_g^{y,t}\|_2) + 2R_f \|\boldsymbol{\mu}_g^{y,t+1} - \boldsymbol{\mu}_g^{y,t}\|_2 \Big] \\
&\overset{(3)}{\leq} \alpha \cdot \max_y \|\boldsymbol{\mu}_g^{y,t+1} - \boldsymbol{\mu}_g^{y,t}\|_2 \cdot (2R_f + 2R_\mu) \\
&\overset{(4)}{\leq} 2\alpha(R_f + R_\mu) \sum_{j=1}^{m} w_j \max_y \|\boldsymbol{\mu}_j^{y,t} - \boldsymbol{\mu}_j^{y,t-1}\|_2 \\
&\overset{(5)}{\leq} 2\alpha L_2(R_f + R_\mu) \sum_{j=1}^{m} w_j \mathbb{E}\Big[ \|\phi_j^{E,t} - \phi_j^{\frac{1}{2},t}\|_2 \Big] \cdot M_\mu \\
&\overset{(6)}{\leq} 4\alpha L_2(R_f + R_\mu) \eta_0 EV M_\mu,
\end{aligned}
\tag{74}
$$

where (2) follows from triangle inequality and Cauchy-Schwarz inequality; (3) follows from Assumption G.5 (Equation (45)); (4) follows from aggregation rule (Equation (31)); (5) follows from Assumption G.1 and Assumption G.4; (6) follows from $\mathbb{E}[\|\phi_j^{E,t} - \phi_j^{\frac{1}{2},t}\|] \leq \eta_0 EV$.

*Term 3: Variance alignment loss.*

The change in variance alignment loss is:

$$\Delta\mathcal{L}_{\text{var}} = \frac{\gamma}{|\mathcal{C}_i \cap \mathcal{C}_g|} \sum_{c \in \mathcal{C}_i \cap \mathcal{C}_g} \left[ \|\boldsymbol{\sigma}_i^c - \boldsymbol{\sigma}_g^{c,t+1}\|_2^2 - \|\boldsymbol{\sigma}_i^c - \boldsymbol{\sigma}_g^{c,t}\|_2^2 \right]. \tag{75}$$

$$\Delta\mathcal{L}_{\text{var}} \overset{(1)}{=} \frac{\gamma}{|\mathcal{C}_i \cap \mathcal{C}_g|} \sum_{c \in \mathcal{C}_i \cap \mathcal{C}_g} \left[ \|\boldsymbol{\sigma}_g^{c,t+1}\|_2^2 - \|\boldsymbol{\sigma}_g^{c,t}\|_2^2 - 2\langle \boldsymbol{\sigma}_i^c, \boldsymbol{\sigma}_g^{c,t+1} - \boldsymbol{\sigma}_g^{c,t} \rangle \right], \tag{76}$$

where (1) follows from the same identity as in Term 3.

Taking absolute value:

$$
\begin{aligned}
|\Delta\mathcal{L}_{\text{var}}| &\overset{(2)}{\leq} \frac{\gamma}{|\mathcal{C}_i \cap \mathcal{C}_g|} \sum_{c \in \mathcal{C}_i \cap \mathcal{C}_g} \|\boldsymbol{\sigma}_g^{c,t+1} - \boldsymbol{\sigma}_g^{c,t}\|_2 \cdot (2\|\boldsymbol{\sigma}_i^c\|_2 + \|\boldsymbol{\sigma}_g^{c,t+1}\|_2 + \|\boldsymbol{\sigma}_g^{c,t}\|_2) \\
&\overset{(3)}{\leq} \frac{4\gamma M_\sigma}{|\mathcal{C}_i \cap \mathcal{C}_g|} \sum_{c \in \mathcal{C}_i \cap \mathcal{C}_g} \sum_{j=1}^{m} w_j \|\boldsymbol{\sigma}_j^{c,t} - \boldsymbol{\sigma}_j^{c,t-1}\|_2 \\
&\overset{(4)}{\leq} \frac{4\gamma L_2 M_\sigma}{|\mathcal{C}_i \cap \mathcal{C}_g|} \sum_{c \in \mathcal{C}_i \cap \mathcal{C}_g} \sum_{j=1}^{m} w_j \mathbb{E}\left[ \|\phi_j^{E,t} - \phi_j^{\frac{1}{2},t}\|_2 \right] \\
&\overset{(5)}{=} 4\gamma L_2 M_\sigma \sum_{j=1}^{m} w_j \mathbb{E}\left[ \|\phi_j^{E,t} - \phi_j^{\frac{1}{2},t}\|_2 \right] \\
&\overset{(6)}{\leq} 4\gamma L_2 \eta_0 E V M_\sigma,
\end{aligned}
\tag{77}
$$

where (2) follows from triangle inequality and Cauchy-Schwarz inequality; (3) follows from Assumption G.4 (Equation (42)); (4) follows from Assumption G.1 (Equation (35)); (5) follows from canceling the normalization factor $\frac{1}{|\mathcal{C}_i \cap \mathcal{C}_g|}$ with the sum over $|\mathcal{C}_i \cap \mathcal{C}_g|$ terms; (6) follows from $\mathbb{E}[\|\phi_j^{E,t} - \phi_j^{\frac{1}{2},t}\|] \leq \eta_0 E V$.

Combining all terms:

$$
\begin{aligned}
\mathbb{E}\left[ \mathcal{L}_i^{\frac{1}{2},(t+1)} \right] \leq \mathcal{L}_i^{E,t} &+ 2\eta_0^2 L_1 E^2 V^2 + \xi C_{\text{KL}} L_2 L_3 \eta_0 E V + \xi C_{\text{KL}} M_p \\
&+ 4\alpha L_2 (R_f + R_\mu)\eta_0 E V M_\mu + 4\gamma L_2 \eta_0 E V M_\sigma,
\end{aligned}
\tag{78}
$$

which completes the proof of Lemma G.8. $\qquad\square$

## G.3. Theorems

**Theorem G.9** (FedCDWA Convergence). *Under Assumptions G.1–G.5, for an arbitrary client $i$, after each communication round the loss function is bounded as:*

$$
\begin{aligned}
\mathbb{E}\left[ \mathcal{L}_i^{\frac{1}{2},t+1} \right] \leq \mathcal{L}_i^{\frac{1}{2},t} &- \sum_{e=\frac{1}{2}}^{E-1} \left( \eta_e - \frac{\eta_e^2 L_1}{2} \right) \|\nabla \mathcal{L}^{e,t}\|_2^2 \\
&+ 2\eta_0^2 L_1 E^2 V^2 + \frac{\eta_0^2 L_1 E}{2} \sigma^2 + C_{KD} + C_{proto} + C_{var},
\end{aligned}
\tag{79}
$$

*where each term corresponds to $C_{KD}$, $C_{proto}$, and $C_{var}$ as defined in Lemma G.8.*

*Proof.* Combining Lemmas G.7 and G.8, we have:

$$
\begin{aligned}
\mathbb{E}\left[\mathcal{L}_i^{\frac{1}{2},t+1}\right] &= \mathbb{E}\left[\mathcal{L}_i^{E,t} + (\mathcal{L}_i^{\frac{1}{2},t+1} - \mathcal{L}_i^{E,t})\right] \\
&\le \mathcal{L}_i^{E,t} + 2\eta_0^2 L_1 E^2 V^2 + C_{\text{total}} \\
&\le \mathcal{L}_i^{\frac{1}{2},t} - \sum_{e=\frac{1}{2}}^{E-1}\left(\eta_e - \frac{\eta_e^2 L_1}{2}\right)\|\nabla\mathcal{L}^{e,t}\|_2^2 + \frac{\eta_0^2 L_1 E}{2}\sigma^2 \\
&\quad + 2\eta_0^2 L_1 E^2 V^2 + C_{\text{total}}.
\end{aligned}
\tag{80}
$$

where $C_{\text{total}} = C_{\text{KD}} + C_{\text{proto}} + C_{\text{var}}$. $\qquad\square$

**Corollary G.10** (Convergence Condition). *For the loss to decrease in expectation ($\mathbb{E}[\mathcal{L}_i^{\frac{1}{2},t+1}] < \mathcal{L}_i^{\frac{1}{2},t}$), the learning rate must satisfy:*

$$
\eta_0 < \min\left\{\frac{2}{L_1}, \frac{1}{L_1} \cdot \frac{\bar{G}^2 - C_{FedCDWA}}{EV^2 + \sigma^2}\right\},
\tag{81}
$$

*where $\bar{G} = \frac{1}{E}\sum_{e=\frac{1}{2}}^{E-1}\|\nabla\mathcal{L}^{e,t}\|_2$ is the average gradient norm, and*

$$
C_{FedCDWA} = \frac{2C_{total}}{E\eta_0 L_1} = \frac{2\xi C_{KL}L_2 L_3 V}{L_1} + \frac{2\xi C_{KL}M_p}{E\eta_0 L_1} + \frac{8\alpha L_2(R_f + R_\mu)VM_\mu}{L_1} + \frac{8\gamma L_2 VM_\sigma}{L_1}.
\tag{82}
$$

*Proof.* For the loss to decrease, we need:

$$
\sum_{e=\frac{1}{2}}^{E-1}\left(\eta_0 - \frac{\eta_0^2 L_1}{2}\right)\|\nabla\mathcal{L}^{e,t}\|_2^2 > \frac{\eta_0^2 L_1 E}{2}(EV^2 + \sigma^2) + C_{\text{total}}.
\tag{83}
$$

Condition 1: For the coefficient $(\eta_0 - \frac{\eta_0^2 L_1}{2})$ to be positive, we need:

$$
\eta_0 - \frac{\eta_0^2 L_1}{2} > 0 \quad \Rightarrow \quad \eta_0\left(1 - \frac{\eta_0 L_1}{2}\right) > 0 \quad \Rightarrow \quad \eta_0 < \frac{2}{L_1}.
\tag{84}
$$

Condition 2: Assuming constant learning rate $\eta_e = \eta_0$ and dividing the convergence inequality by $E$:

$$
\left(\eta_0 - \frac{\eta_0^2 L_1}{2}\right)\bar{G}^2 > \frac{\eta_0^2 L_1}{2}(EV^2 + \sigma^2) + \frac{C_{\text{total}}}{E},
\tag{85}
$$

where $\bar{G}^2 = \frac{1}{E}\sum_e \|\nabla\mathcal{L}^{e,t}\|_2^2$.

Rearranging:

$$
\eta_0\bar{G}^2 > \frac{\eta_0^2 L_1}{2}(\bar{G}^2 + EV^2 + \sigma^2) + \frac{C_{\text{total}}}{E}.
\tag{86}
$$

When $\eta_0$ is sufficiently small, the condition becomes $\eta_0 < \frac{1}{L_1} \cdot \frac{\bar{G}^2 - C_{\text{FedCDWA}}}{EV^2+\sigma^2}$, which yields the condition in Equation (81). $\quad\square$

**Theorem G.11** (FedCDWA Convergence Rate). *Under Assumptions G.1–G.5, if the learning rate is set to $\eta_e = \eta_0$ for all epochs, and defining $\Delta = \mathcal{L}^{\frac{1}{2},1} - \mathcal{L}^*$ where $\mathcal{L}^*$ is the optimal loss, after $T$ rounds, it holds that:*

$$
\frac{1}{TE}\sum_{t=1}^{T}\sum_{e=\frac{1}{2}}^{E-1}\|\nabla\mathcal{L}^{e,t}\|_2^2 \le \epsilon,
\tag{87}
$$

*If the total number of rounds satisfies:*

$$
T \ge \frac{2\Delta}{\epsilon E(2\eta_0 - \eta_0^2 L_1) - 4\eta_0^2 L_1 E^2 V^2 - \eta_0^2 L_1 E\sigma^2 - 2C_{total}}.
\tag{88}
$$

*Proof.* From Theorem G.9, summing over $T$ rounds:

$$\sum_{t=1}^{T}\sum_{e=\frac{1}{2}}^{E-1}\left(\eta_0 - \frac{\eta_0^2 L_1}{2}\right)\|\nabla\mathcal{L}^{e,t}\|_2^2$$

$$\leq \sum_{t=1}^{T}\left[\mathcal{L}_i^{\frac{1}{2},t} - \mathbb{E}\left[\mathcal{L}_i^{\frac{1}{2},t+1}\right]\right]$$

$$+ T\left(2\eta_0^2 L_1 E^2 V^2 + \frac{\eta_0^2 L_1 E}{2}\sigma^2 + \xi C_{\text{KL}} L_2 L_3 \eta_0 EV + \xi C_{\text{KL}} M_p\right.$$

$$\left. + 4\alpha L_2(R_f + R_\mu)\eta_0 EV M_\mu + 4\gamma L_2 \eta_0 EV M_\sigma\right)$$

$$\leq \mathcal{L}_i^{\frac{1}{2},1} - \mathbb{E}\left[\mathcal{L}_i^{\frac{1}{2},T+1}\right]$$

$$+ T\left(2\eta_0^2 L_1 E^2 V^2 + \frac{\eta_0^2 L_1 E}{2}\sigma^2 + \xi C_{\text{KL}} L_2 L_3 \eta_0 EV + \xi C_{\text{KL}} M_p\right.$$

$$\left. + 4\alpha L_2(R_f + R_\mu)\eta_0 EV M_\mu + 4\gamma L_2 \eta_0 EV M_\sigma\right)$$

$$\leq \Delta + T\left(2\eta_0^2 L_1 E^2 V^2 + \frac{\eta_0^2 L_1 E}{2}\sigma^2 + \xi C_{\text{KL}} L_2 L_3 \eta_0 EV + \xi C_{\text{KL}} M_p\right.$$

$$\left. + 4\alpha L_2(R_f + R_\mu)\eta_0 EV M_\mu + 4\gamma L_2 \eta_0 EV M_\sigma\right). \tag{89}$$

Dividing both sides by $TE\left(\eta_0 - \frac{\eta_0^2 L_1}{2}\right)$ and rearranging:

$$\frac{1}{TE}\sum_{t=1}^{T}\sum_{e=\frac{1}{2}}^{E-1}\|\nabla\mathcal{L}^{e,t}\|_2^2 \leq \frac{\Delta}{TE\left(\eta_0 - \frac{\eta_0^2 L_1}{2}\right)}$$

$$+ \frac{1}{E\left(\eta_0 - \frac{\eta_0^2 L_1}{2}\right)}\left(2\eta_0^2 L_1 E^2 V^2 + \frac{\eta_0^2 L_1 E}{2}\sigma^2\right.$$

$$+ \xi C_{\text{KL}} L_2 L_3 \eta_0 EV + \xi C_{\text{KL}} M_p$$

$$\left. + 4\alpha L_2(R_f + R_\mu)\eta_0 EV M_\mu + 4\gamma L_2 \eta_0 EV M_\sigma\right). \tag{90}$$

Setting the right-hand side to $\epsilon$ yields:

$$\frac{1}{TE}\sum_{t=1}^{T}\sum_{e=\frac{1}{2}}^{E-1}\|\nabla\mathcal{L}^{e,t}\|_2^2 \leq \frac{\Delta}{TE\left(\eta_0 - \frac{\eta_0^2 L_1}{2}\right)}$$

$$+ \frac{1}{E\left(\eta_0 - \frac{\eta_0^2 L_1}{2}\right)}\left(2\eta_0^2 L_1 E^2 V^2 + \frac{\eta_0^2 L_1 E}{2}\sigma^2\right.$$

$$+ \xi C_{\text{KL}} L_2 L_3 \eta_0 EV + \xi C_{\text{KL}} M_p$$

$$\left. + 4\alpha L_2(R_f + R_\mu)\eta_0 EV M_\mu + 4\gamma L_2 \eta_0 EV M_\sigma\right) \leq \epsilon. \tag{91}$$

Therefore,

$$\frac{\Delta}{T} \leq \epsilon E\left(\eta_0 - \frac{\eta_0^2 L_1}{2}\right) - 2\eta_0^2 L_1 E^2 V^2 - \frac{\eta_0^2 L_1 E}{2}\sigma^2$$

$$- \xi C_{\text{KL}} L_2 L_3 \eta_0 EV - \xi C_{\text{KL}} M_p$$

$$- 4\alpha L_2(R_f + R_\mu)\eta_0 EV M_\mu - 4\gamma L_2 \eta_0 EV M_\sigma. \tag{92}$$

Rearranging yields the convergence rate:

$$T \geq \frac{2\Delta}{\epsilon E(2\eta_0 - \eta_0^2 L_1) - 4\eta_0^2 L_1 E^2 V^2 - \eta_0^2 L_1 E\sigma^2 - 2C_{\text{total}}}. \tag{93}$$

This completes the proof. $\qquad\square$

