# OpenReview forum: "FedCDWA: Decoupled Federated Prototype Distillation with Hierarchical Wasserstein Aggregation"
_ICML.cc/2026/Conference — ICML 2026 regular_

### Official Review · Reviewer_VqoY · 2026-03-11

**Soundness:** 2
**Presentation:** 3
**Significance:** 2
**Originality:** 3
**Overall Recommendation:** 4
**Confidence:** 4

**Summary:**

This paper addresses the challenge of data heterogeneity in FL by propsing the FedCDWA, which consists of a Decoupled Distillation Strategy (DDS) separates client-side personalized distillation from server-side mutual distillation to resolve optimization conflicts, a Hierarchical Wasserstein Aggregation (HWA) mechanism uses optimal transport theory to aggregate prototypes both within and across classes, and a Prototype-Variance Dual Alignment (PVDA) mechanism aligns both first and second order statistics for fine-grained distribution matching. Experiments have been performed on 3 datasets, including SVHN, CIFAR-10, and CIFAR-100, and the proposed method outperforms compared methods in both local and global accuracy.

**Compliance With Llm Reviewing Policy:**

Affirmed.

**Final Justification:**

My concerns have been addressed, and I will keep my current positive rating.

**Key Questions For Authors:**

Please see the weakness part.

**Limitations:**

yes

**Strengths And Weaknesses:**

# Strength
- Well-motivated and clear problem identification
- Paper is well organized and easy to follow
- The proposed method outperforms compared methods and shows consistent performance improvements.

# Weakness

- A more detailed analysis regarding the computational efficiency is needed.  With increased number of classes, the overhead becomes high. The total time reported in Table 1 aggregates client training, communication, and server-side aggregation. It is unclear what portion of this time is solely attributable to the proposed components.

- The PVDA mechanism aligns local variance with the global. However, this may raise concerns in highly heterogeneous settings where a client may hold an imbalanced dataset, and there are very few samples certain class, the estimate of local variance could be unreliable.

- The experiments are mainly evaluated under label distribution skew. However, the heterogeneity also includes the feature distribution skew, the authors should clearly define the scope, or extend the validation to feature skew.

- The feature dimension is fixed relatively low. It's unclear how the framework scales and performs with higher-dimensional features.

- The potential privacy leakage has not been discussed. Sharing second-order statistics provides more information than means.

---

> ### Author Rebuttal · Authors · 2026-03-31
>
> ## For W1:
> Thank you for this reminder. **The detailed discussion is included in our response to reviewer pmXQ for W2 & Q1**. It is worth noting that Table 1 reports the average per-round wall-clock time per client. This metric can reflect the runtime cost of client local training. Although it cannot directly isolate the individual cost of each proposed component, it also suggests that, in FedCDWA, there is likewise a trade-off between client-side local training cost and local accuracy.
> ## For W2:
> We agree that this is a valid edge-case concern under stronger heterogeneity and low-sample client-class conditions. **Our main experiments follow the same Dirichlet partition method [1-3] as prior prototype/distillation-based FL works and are not an extreme degenerate setting. To test this concern, we implemented a sample-count-aware weighted variance alignment variant under a more heterogeneous setting (CIFAR-100, 50 clients,$\beta$=0.2), with reliability weight**
> $$r_i^c =
> \frac{|F_i^c|}{|F_i^c| + \tau_{var}}.
> $$
>
> Then rewrite the variance alignment term as
> $$L_{var}^{weight}=
>  \gamma \cdot \frac{1}{\sum_{c \in C_i \cap C_g} r_i^c}
> \sum_{c \in C_i \cap C_g}
> r_i^c \lVert \sigma_i^c - \sigma_g^c \rVert_2^2.
> $$
>
> **Table A. Robustness check of weighted variance alignment under $\beta = 0.2$**
> | Dataset | Method | $\tau_{var}$ | Local Acc (%) | Global Acc (%) | $\Delta$Local | $\Delta$Global |
> |---|---|---|---|---|---|---|
> | CIFAR-100 | FedCDWA | - | 34.63 | 26.06 | - | - |
> | CIFAR-100 | FedCDWA w/ weighted | 5 | 33.47 | 26.81 | -1.16 | +0.75 |
> | CIFAR-100 | FedCDWA w/ weighted | 10 | 32.91 | 25.67 | -1.66 | -0.39 |
>
> The results in Table A show that mild reliability-aware down weighting can slightly improve global performance ($\tau_{var}=5$), while an overly conservative choice ($\tau_{var}=10$) instead degrades performance. This suggests that the reviewer’s concern is meaningful, but the original PVDA already exhibits a certain degree of robustness, and moderate rather than overly strong reliability control is more appropriate for low-sample classes.
> ## For W3:
> We agree with the reviewer’s reminder. The current empirical study mainly focuses on label-distribution heterogeneity: Table 1 and the appendix experiments with different $\beta$ values are based on Dirichlet non-IID partitioning. At the same time, FedCDWA is not limited to label skew: both HWA and PVDA operate directly in feature space to alleviate geometric distortion and fine-grained distribution mismatch. Thus, the framework is also theoretically well motivated for feature-skew. scenarios, while dedicated feature-skew benchmarks are an important direction for future work.
> ## For W4:
> Thank you for raising this issue. We further increased the feature dimension from 32 to 64 under the same 50-client setting as the main experiments. The results in Table B show that FedCDWA remains stable and competitive in a higher-dimensional feature space.
>
> **Table B. Performance with higher-dimensional feature dimension**
> | Method | Local Acc | Global Acc |
> | --- | --- | --- |
> | Dataset: SVHN |  |  |
> | FedAvg | 84.07 | 90.37 |
> | FedProx | 86.54 | 90.52 |
> | Moon | 86.38 | 90.29 |
> | FedMD | 86.03 | 85.75 |
> | FedGen | 85.49 | 89.89 |
> | FedProto | 85.84 | 85.87 |
> | FedHKD | 88.31 | 90.75 |
> | FPL | 85.13 | 89.72 |
> | FedGMKD | 88.38 | 90.81 |
> | FedCDWA | **88.78** | **91.72** |
> | Dataset: CIFAR-10 |  |  |
> | FedAvg | 62.77 | 58.83 |
> | FedProx | 64.87 | 59.01 |
> | Moon | 63.99 | 58.60 |
> | FedMD | 63.21 | 48.24 |
> | FedGen | 58.40 | 57.83 |
> | FedProto | 57.81 | 50.98 |
> | FedHKD | 64.23 | 60.09 |
> | FPL | 63.55 | 60.33 |
> | FedGMKD | 63.41 | 60.21 |
> | FedCDWA | **66.18** | **61.43** |
> | Dataset: CIFAR-100 |  |  |
> | FedAvg | 24.30 | 26.70 |
> | FedProx | 25.26 | 28.54 |
> | Moon | 23.16 | 25.89 |
> | FedMD | 26.54 | 28.15 |
> | FedGen | 24.72 | 26.55 |
> | FedProto | 24.96 | 28.39 |
> | FedHKD | 26.78 | 29.09 |
> | FPL | 23.35 | 24.86 |
> | FedGMKD | 24.91 | 29.44 |
> | FedCDWA | **27.22** | **30.05** |
>
> ## For W5:
> This is a reasonable and important concern. The privacy issue has also been discussed in our response to reviewer 8KXq for W3&Q2. In addition, compared with the methods discussed above, FedCDWA introduces variance alignment to enhance fine-grained distribution alignment. Therefore, we agree that such statistics indeed contain richer distributional information. Based on this point, we will add to the limitations discussion in the revised version that FedCDWA can be combined with mechanisms such as differential privacy in future work to further protect these exchanged statistical information.
> ## References:
> [1] FedProto: Federated Prototype Learning across Heterogeneous Clients. AAAI, 2022.
> [2] The Best of Both Worlds: Accurate Global and Personalized Models through Federated Learning with Data-Free Hyper-Knowledge Distillation. ICLR, 2023.
> [3] FedGMKD: An Efficient Prototype Federated Learning Framework through Knowledge Distillation and Discrepancy-Aware Aggregation. NeurIPS, 2024.

---

> > ### Author Rebuttal · Reviewer_VqoY · 2026-04-03
> >
> > Thanks for the authors detailed response. I have no follow-up questions and happy to keep my current score.

---

> > > ### Author Response · Authors · 2026-04-03
> > >
> > > We sincerely appreciate your acknowledgment of our response and are grateful for your constructive feedback throughout the review process. Thank you again for your time and consideration.

---

### Official Review · Reviewer_8KXq · 2026-03-11

**Soundness:** 2
**Presentation:** 2
**Significance:** 2
**Originality:** 2
**Overall Recommendation:** 3
**Confidence:** 2

**Summary:**

This paper presents FedCDWA, a federated learning algorithm that aims at mitigating the impact of client data heterogeneity. It combines three different techniques: (i) a decoupled distillation strategy; (ii) a hierarchical Wasserstein aggregation; and (iii) a prototype variance dual alignment mechanism. The evaluation considers three image datasets: SVHN, CIFAR-10 and CIFAR-100.

**Compliance With Llm Reviewing Policy:**

Affirmed.

**Key Questions For Authors:**

* Is the code available?
* Is it not a privacy issue to share per-client statistics (Sec. 3.3.1)? Or am I misunderstanding something?
* Can the intuition behind the use of these three building blocks be summarized?

**Limitations:**

yes

**Strengths And Weaknesses:**

Strengths
* Classical problem.
* There is a convergence analysis.
* Experiments on three datasets with 9 relevant baselines indicate superior performance.

Weaknesses
* The topic is well known and not really original.
* The paper relies too much on its (long) appendix. The main body of the paper does not contain pseudocode, and does not explain well the rationales behind FedCDWA’s design. The motivation and intuition for these building blocks are not explicitly stated.
* The hierarchical Wasserstein aggregation seems to make use of per-client class statistics. I think that sharing those statistics is generally avoided for privacy reasons.
* There are missing white spaces before many references.
* Experiments are done with local simulations that do not involve real communications between the server and the clients.
* The paragraph on computational efficiency does not report any number.
* From the result table it is not clear to me whether FedCDWA clearly significantly outperforms other baselines.
* The code does not seem to have been released

---

> ### Author Rebuttal · Authors · 2026-03-31
>
> ## For W1:
>
> Thank you for the assessment. Federated learning under heterogeneous data is an important problem that has been widely studied, and prototype/distillation-based FL has also attracted substantial attention in a number of representative works. Our contribution is not a new topic per se, but a targeted treatment of client drift in federated prototype distillation under highly heterogeneous client data.
>
> ## For W2 & Q3:
>
> Thank you for the suggestion. **The complete algorithmic pseudocode is indeed provided in Appendix A, and line 135 in the manuscript explicitly points to Appendix A. At the same time, the design motivation of FedCDWA is already outlined in Figure 1 and the Introduction.** Specifically, we focus on the client-drift problem caused by heterogeneous data in federated prototype distillation. The three building blocks are designed to address three key challenges, as follows:
>
> - **DDS** mitigates optimization conflicts in coupled distillation. In bidirectional distillation, client and server can form non-stationary teacher-student feedback within the same round. DDS decouples client-side personalized prototype distillation from server-side mutual distillation, so that clients absorb fixed global knowledge and the server refines the global model afterwards in a unified way.
> - **HWA** alleviates geometric distortion caused by simple prototype averaging. Under strong heterogeneity, direct averaging can be biased toward dominant clients and distort the global feature geometry. HWA therefore uses intra-class Wasserstein aggregation and inter-class Sliced Wasserstein refinement to better preserve class relationships.
> - **PVDA** aligns both first- and second-order statistics to reduce the mismatch left by center-only alignment. Even if class means are close across clients, intra-class dispersion and variance structure may still differ (“same center, different shape”). PVDA addresses this through finer-grained statistical alignment.
>
> ## For W3 & Q2:
>
> This is a reasonable concern. Similar to existing prototype/distillation-based FL methods [2-4], our current goal is to improve prototype-distillation performance rather than provide formal privacy guarantees for exchanged statistics. This issue is orthogonal to privacy-preserving research. We will define this scope more clearly in the revised version and add a corresponding discussion in the limitations section.
>
> ## For W4:
>
> Thank you for pointing out this formatting issue. We will carefully proofread and correct the missing spaces before references and related typesetting issues in the revision.
>
> ## For W5:
>
> Our experiments are conducted in a controlled simulated cross-device setting rather than a real-world cross-device deployment. This follows the standard protocol of prior FL algorithm [1- 4] papers, whose focus is algorithm-level optimization/generalization under heterogeneity rather than system-level communication evaluation.
>
> ## For W6:
>
> We would like to clarify that Table 1 already reports the computational-efficiency numbers: the average per-round wall-clock time per client of FedCDWA is 2.49s / 3.35s / 3.39s on SVHN / CIFAR-10 / CIFAR-100, measured on a single RTX 3090 under parallel execution. As discussed in Sec. 4.4 (lines 427- 438), the overhead is nearly the same as FedAvg on SVHN and moderately higher on CIFAR-10/CIFAR-100. If the reviewer’s concern is that these numbers are not explicitly restated in the paragraph of Sec. 4.4, we will make them more explicit in the revision.
>
> ## For W7:
>
> On this point, we would like to make a clarification. The advantage of FedCDWA has already been presented fairly clearly in Table 1 and Section 4.4 (lines 367-438) of the manuscript: across three datasets and three client-number settings, it consistently outperforms the compared baselines in both local and global accuracy. For example, on CIFAR-10, the manuscript explicitly reports gains over FedAvg of **5.88%–10.50%** in local accuracy and **1.51%–4.83%** in global accuracy. Thus, the superiority of FedCDWA is already supported by stable and consistent experimental evidence.
>
> ## For W8 & Q1:
>
> Thank you for the comment. We are willing, and will certainly, release the code after the paper is accepted, so as to facilitate reproducibility and further research progress on knowledge distillation and prototype learning in heterogeneous FL.
>
> ## Reference
>
> [1] Communication-Efficient Learning of Deep Networks from Decentralized Data. AISTATS, 2017
> [2] FedProto: Federated Prototype Learning across Heterogeneous Clients. AAAI, 2022.
> [3] The Best of Both Worlds: Accurate Global and Personalized Models through Federated Learning with Data-Free Hyper-Knowledge Distillation. ICLR, 2023.
> [4] FedGMKD: An Efficient Prototype Federated Learning Framework through Knowledge Distillation and Discrepancy-Aware Aggregation. NeurIPS, 2024.

---

> > ### Author Rebuttal · Reviewer_8KXq · 2026-04-03
> >
> > Thank you for your answers.

---

> > > ### Author Response · Authors · 2026-04-03
> > >
> > > We sincerely appreciate your acknowledgment of our response and your constructive feedback. If you feel our clarifications improve the overall assessment, we would be grateful if you would consider raising your score.

---

### Official Review · Reviewer_pmXQ · 2026-03-13

**Soundness:** 3
**Presentation:** 3
**Significance:** 3
**Originality:** 3
**Overall Recommendation:** 5
**Confidence:** 3

**Summary:**

This paper addresses the critical problem of "client drift" in Federated Learning caused by highly heterogeneous (Non-IID) client data distributions. The authors identify three major flaws in existing federated prototype distillation methods: (1) gradient conflicts caused by coupled bidirectional distillation; (2) geometric distortion of the global feature space due to simplistic prototype aggregation; and (3) incomplete distribution matching that only aligns first-order centroids. To overcome these, the authors propose FedCDWA, a novel framework featuring three core component, Decoupled Distillation Strategy, Hierarchical Wasserstein Aggregation, and Prototype-Variance Dual Alignment. The paper supports its method with rigorous convergence analysis and extensive experiments on SVHN, CIFAR-10, and CIFAR-100, demonstrating consistent superiority over baselines.

**Compliance With Llm Reviewing Policy:**

Affirmed.

**Final Justification:**

The paper is techinically sound. Although Sliced Wasserstein Distance matrix incurs pairwise computation, the pratical run time of FedCDWA is comeptitive.

**Key Questions For Authors:**

1. How does the computation time of the server-side inter-class Sliced Wasserstein Refinement scale when the number of classes K scales up significantly?
2. Why the average accuracy of SVHN dataset (table 2) is lower than the average accuracy of CIFAR-10 dataset (table3)? This is not consistent with Table 1.

**Limitations:**

yes

**Strengths And Weaknesses:**

### Strengths

- The paper provides a rigorous theoretical convergence analysis mapping out the optimization bounds of the framework
- The empirical performance on SVHN, CIFAR-10, and CIFAR-100 demonstrates clear improvements over existing baselines under various Dirichlet heterogeneity settings.

### Weaknesses
- The framework is highly complex and introduces a multitude of hyperparameters. The ablation studies show that performance is highly sensitive to these parameters.
- While HWA is mathematically sound, the server must compute a pairwise Sliced Wasserstein Distance matrix for all classes to find the top-k similar classes.

---

> ### Author Rebuttal · Authors · 2026-03-31
>
> ## For W1:
>
> Thank you for this important observation. **We emphasize that the best hyperparameter settings (lines 577-603) reported for each baseline when tuning the baselines, and FedCDWA still outperforms the baseline methods across all tested hyperparameter settings; please see Table 1 in the manuscript. In addition, although FedCDWA does require some degree of hyperparameter tuning, it is not highly sensitive to these parameters.** Specifically, Appendix C (line 600) states that $\alpha$ and $\gamma$ are selected by grid search over predefined ranges. Moreover, Tables 7-8 (lines 825-860) and Tables 10-11 (lines 935-951) further show that FedCDWA remains stable over a relatively broad range of hyperparameter values. Even under suboptimal parameter settings, it still outperforms FedAvg. Overall, the results indicate a trade-off across different parameter choices and their corresponding performance, rather than high sensitivity.
>
> ## For W2 & Q1:
>
> Thank you for pointing this out. As shown in Table A and Table B, we provide:
>
> - **Table A. Overall per-round runtime comparison** to compare the **client-side, server-side, and total** time of all methods;
> - **Table B: FedCDWA server-side component runtime**, including **summary aggregation, inter-class SWD refinement, and mutual distillation**, to further decompose the source of its server-side overhead.
>
> **Table A. Overall per-round runtime comparison**
>
> | Method | Client-side | Server-side | Total |
> | --- | :---: | :---: | :---: |
> | **Dataset: SVHN** |  |  |  |
> | FedAvg | 24.87 | 0.02 | 24.89 |
> | FedProx | 39.30 | 0.02 | 39.32 |
> | Moon | 29.61 | 0.51 | 30.12 |
> | FedMD | 44.17 | 0.12 | 44.29 |
> | FedGen | 24.80 | 0.43 | 25.23 |
> | FedProto | 171.05 | 0.02 | 171.07 |
> | FedHKD | 167.33 | 0.02 | 167.35 |
> | FPL | 73.82 | 0.49 | 74.31 |
> | FedGMKD | 200.79 | 0.04 | 200.83 |
> | FedCDWA | 56.81 | 0.10 | 56.91 |
> | **Dataset: CIFAR-10** |  |  |  |
> | FedAvg | 28.17 | 0.02 | 28.19 |
> | FedProx | 48.82 | 0.02 | 48.84 |
> | Moon | 35.96 | 0.50 | 36.46 |
> | FedMD | 49.22 | 0.11 | 49.33 |
> | FedGen | 29.17 | 0.42 | 29.59 |
> | FedProto | 124.74 | 0.02 | 124.76 |
> | FedHKD | 126.00 | 0.02 | 126.02 |
> | FPL | 84.47 | 0.49 | 84.96 |
> | FedGMKD | 149.52 | 0.05 | 149.57 |
> | FedCDWA | 54.72 | 0.11 | 54.83 |
> | **Dataset: CIFAR-100** |  |  |  |
> | FedAvg | 28.29 | 0.02 | 28.31 |
> | FedProx | 48.09 | 0.02 | 48.11 |
> | Moon | 35.54 | 0.51 | 36.05 |
> | FedMD | 48.51 | 0.12 | 48.63 |
> | FedGen | 29.10 | 0.42 | 29.52 |
> | FedProto | 123.88 | 0.02 | 123.90 |
> | FedHKD | 124.83 | 0.05 | 124.88 |
> | FPL | 85.44 | 0.48 | 85.92 |
> | FedGMKD | 147.86 | 0.21 | 148.07 |
> | FedCDWA | 55.29 | 3.25 | 58.54 |
>
> **Table B. FedCDWA server-side component runtime**
>
> | Dataset | Summary aggregation | Inter-class SWD refinement | Mutual distillation | Server-side total |
> |---|---|---|---|---|
> | SVHN | 0.03 | 0.04 | 0.03 | 0.10 |
> | CIFAR-10 | 0.03 | 0.05 | 0.03 | 0.11 |
> | CIFAR-100 | 0.04 | 2.94 | 0.27 | 3.25 |
>
> From Table A, we can see that the total time of FedCDWA on the three datasets is 56.91 / 54.83 / 58.54s, respectively, which remains competitive overall. In particular, compared with FedProto [1], FedHKD [2], FPL [3], and FedGMKD [4], FedCDWA has lower total overhead while achieving better local and global accuracy.
>
> From Table B, we further observe that the main source of the extra overhead is indeed the server-side inter-class SWD refinement: as the number of classes $k$ increases from 10 to 100, this component takes only 0.04s/ 0.05s rises to 2.94s. In a word, FedCDWA incurs only about 3 additional seconds on the server side, the total runtime remains reasonable and acceptable. There is a trade-off between model accuracy and the corresponding time cost.
>
> ## For Q2:
>
> We thank the reviewer for identifying this issue. We confirm that the titles of Table 2 and Table 3 in the appendix (lines 726-757) were accidentally swapped. This was an editing mistake rather than an error in the experimental design, data statistics, or the results themselves. **Importantly, the numerical values in those tables are correct, and the comparative results and quantitative conclusions in Table 1 (lines 385-415) are also correct.** Therefore, this issue does not affect the validity of the experiments or the main conclusions of the paper. We will correct the titles of Appendix Tables 2 and 3 in the revised version. We have also carefully reviewed the remaining figure/table titles, numbering, and confirmed that there are no other inconsistencies of this kind.
>
> ## References:
>
> [1] FedProto: Federated prototype learning across heterogeneous clients. AAAI, 2022.
> [2] The best of both worlds: Accurate global and personalized models through federated learning with data-free hyper-knowledge distillation. ICLR, 2023.
> [3] Rethinking federated learning with domain shift: A prototype view. CVPR, 2023
> [4] FedGMKD: An efficient prototype federated learning framework through knowledge distillation and discrepancy-aware aggregation. NeurIPS, 2024.

---

> > ### Author Rebuttal · Reviewer_pmXQ · 2026-04-02
> >
> > I thank the authors for the detailed rebuttal. I have no further questions.

---

> > > ### Author Response · Authors · 2026-04-02
> > >
> > > We sincerely appreciate your acknowledgment of our efforts to address your concerns, and we thank you for your constructive feedback throughout the review process.

---

### Decision · Program_Chairs · 2026-04-30

**Decision:**

Accept (regular)

**Comment:**

This paper develops a new method for federated learning under data heterogeneity via prototype distillation with hierarchical Wasserstein aggregation. The high-level idea is to decouple client-side personalized distillation from server-side distillation.

Overall, the reviewers are positive with the contribution and find the rebuttal satisfactory. Given this, I will recommend acceptance. However, I would like to bring to the authors' attention that this high-level tactic has been explored before and the authors appear to miss recent literature addressing this challenge.

[1] https://arxiv.org/abs/2502.19752 (NeurIPS-24)
[2] https://arxiv.org/abs/2510.22880 (NeurIPS-25)
[3] https://arxiv.org/abs/2602.07081 (ICCV-25)

While the technical methods are different, the proposed solutions all aim to solve the same or similar problems (on federated data heterogeneity) via client-side distillation into prompt sets which is decoupled from server-side aggregation. There should have been at least some discussion with such directly relevant works.

In this case, I will not penalize the paper for missing those related papers because they are fairly recent (though the NeurIPS-24 paper is not so recent) but I'd like to remind the authors to be more exhaustive in scanning recent literature. Otherwise, we will risk fragmenting the literature. Please consider update the literature review to directly discuss and position the proposed methods with those relevant works.